# Learning to summarize from human feedback

**Nisan Stiennon**[*]    **Long Ouyang**[*]    **Jeff Wu**[*]    **Daniel M. Ziegler**[*]    **Ryan Lowe**[*]

**Chelsea Voss**[*]      **Alec Radford**      **Dario Amodei**      **Paul Christiano**[*]

OpenAI

## Abstract

As language models become more powerful, training and evaluation are increasingly bottlenecked by the data and metrics used for a particular task. For example, summarization models are often trained to predict human reference summaries and evaluated using ROUGE, but both of these metrics are rough proxies for what we really care about—summary quality. In this work, we show that it is possible to significantly improve summary quality by training a model to optimize for human preferences. We collect a large, high-quality dataset of human comparisons between summaries, train a model to predict the human-preferred summary, and use that model as a reward function to fine-tune a summarization policy using reinforcement learning. We apply our method to a version of the TL;DR dataset of Reddit posts [63] and find that our models significantly outperform both human reference summaries and much larger models fine-tuned with supervised learning alone. Our models also transfer to CNN/DM news articles [22], producing summaries nearly as good as the human reference without any news-specific fine-tuning.[2] We conduct extensive analyses to understand our human feedback dataset and fine-tuned models.[3] We establish that our reward model generalizes to new datasets, and that optimizing our reward model results in better summaries than optimizing ROUGE according to humans. We hope the evidence from our paper motivates machine learning researchers to pay closer attention to how their training loss affects the model behavior they actually want.

## 1 Introduction

Large-scale language model pretraining has become increasingly prevalent for achieving high performance on a variety of natural language processing (NLP) tasks. When applying these models to a specific task, they are usually fine-tuned using supervised learning, often to maximize the log probability of a set of human demonstrations.

While this strategy has led to markedly improved performance, there is still a misalignment between this fine-tuning objective—maximizing the likelihood of human-written text—and what we care about—generating high-quality outputs as determined by humans. This misalignment has several causes: the maximum likelihood objective has no distinction between important errors (e.g. making up facts [41]) and unimportant errors (e.g. selecting the precise word from a set of synonyms); models

---

[*]This was a joint project of the OpenAI Reflection team. Author order was randomized amongst {LO, JW, DZ, NS}; CV and RL were full-time contributors for most of the duration. PC is the team lead.

[2]Samples from all of our models can be viewed on our website.

[3]We provide inference code for our 1.3B models and baselines, as well as a model card and our human feedback dataset with over 64k summary comparisons, here.

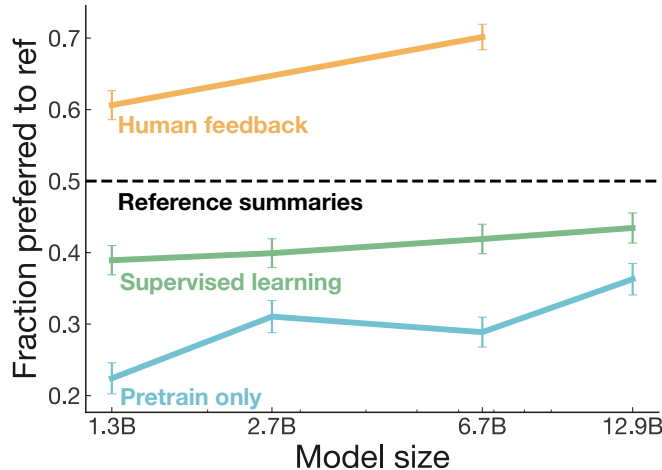

Figure 1: Fraction of the time humans prefer our models' summaries over the human-generated reference summaries on the TL;DR dataset.[4]Since quality judgments involve an arbitrary decision about how to trade off summary length vs. coverage within the 24-48 token limit, we also provide length-controlled graphs in Appendix F; length differences explain about a third of the gap between feedback and supervised learning at 6.7B.

are incentivized to place probability mass on all human demonstrations, including those that are low-quality; and distributional shift during sampling can degrade performance [56, 52]. Quality can often be improved significantly by non-uniform sampling strategies such as beam search [51], but these can lead to repetition and other undesirable artifacts [69, 23]. Optimizing for quality may be a principled approach to overcoming these problems.

Our goal in this paper is to advance methods for training language models on objectives that more closely capture the behavior we care about. To make short-term progress towards this goal, we focus on abstractive English text summarization, as it has a long history in the NLP community [16, 8, 54, 59, 50], and is a subjective task where we believe it is difficult to quantify summary quality without human judgments. Indeed, existing automatic metrics for evaluating summary quality, such as ROUGE [39], have received criticism for poor correlation with human judgments [55, 45, 6, 33].

We follow the works of [3, 73], who fine-tune language models from human feedback using reward learning [35]. We first collect a dataset of human preferences between pairs of summaries, then train a reward model (RM) via supervised learning to predict the human-preferred summary. Finally, we train a policy via reinforcement learning (RL) to maximize the score given by the RM; the policy generates a token of text at each 'time step', and is updated using the PPO algorithm [58] based on the RM 'reward' given to the entire generated summary. We can then gather more human data using samples from the resulting policy, and repeat the process. We follow the works of [48, 4] and use large pretrained GPT-3 models with as many as 6.7 billion parameters.

Our main contributions are four-fold.

**(1) We show that training with human feedback significantly outperforms very strong baselines on English summarization.** When applying our methods on a version of the Reddit TL;DR dataset [63], we train policies via human feedback that produce better summaries than much larger policies trained via supervised learning. Summaries from our human feedback models are preferred by our labelers to the original human demonstrations in the dataset (see Figure 1).

**(2) We show human feedback models generalize much better to new domains than supervised models.** Our Reddit-trained human feedback models also generate high-quality summaries of news articles on the CNN/DailyMail (CNN/DM) dataset without any news-specific fine-tuning, almost matching the quality of the dataset's reference summaries. We perform several checks to ensure that these human preferences reflect a real quality difference: we consistently monitor agreement rates amongst labelers and researchers, and find researcher-labeler agreement rates are nearly as high as researcher-researcher agreement rates (see Section C.2), and we verify models are not merely optimizing simple metrics like length or amount of copying (see Appendices F and G.7).

**(3) We conduct extensive empirical analyses of our policy and reward model.** We examine the impact of model and data size (Figure 6), study performance as we continue to optimize a given reward model (Section 4.3), and analyze reward model performance using synthetic and human-written perturbations of summaries (Section 4.3). We confirm that our reward model outperforms other metrics such as ROUGE at predicting human preferences, and that optimizing our reward model directly results in better summaries than optimizing ROUGE according to humans (Section 4.4).

**(4) We publicly release our human feedback dataset for further research.** The dataset contains 64,832 summary comparisons on the TL;DR dataset, as well as our evaluation data on both TL;DR (comparisons and Likert scores) and CNN/DM (Likert scores).

The methods we present in this paper are motivated in part by longer-term concerns about the misalignment of AI systems with what humans want them to do. When misaligned summarization models make up facts, their mistakes are fairly low-risk and easy to spot. However, as AI systems become more powerful and are given increasingly important tasks, the mistakes they make will likely become more subtle and safety-critical, making this an important area for further research.

## 2    Related work

Most directly related to our work is previous work using human feedback to train summarization models with RL [3, 73]. Bohm et al. [3] learn a reward function from a dataset of human ratings of 2.5k CNN/DM summaries, and train a policy whose summaries are preferred to a policy optimizing ROUGE. Our work is most similar to [73], who also train Transformer models [62] to optimize human feedback across a range of tasks, including summarization on the Reddit TL;DR and CNN/DM datasets. Unlike us, they train in an online manner and find the model highly extractive. They note that their labelers prefer extractive summaries and have low agreement rates with researchers. Compared to [73], we use significantly larger models, move to the batch setting for collecting human feedback, ensure high labeler-researcher agreement, and make some algorithmic modifications, such as separating the policy and value networks.

Human feedback has also been used as a reward to train models in other domains such as dialogue [25, 68, 21], translation [32, 1], semantic parsing [34], story generation [72], review generation [7], and evidence extraction [46]. Our reward modeling approach was developed in prior work on learning to rank [40], which has been applied to ranking search results using either explicit feedback [2, 18] or implicit feedback in the form of click-through data [29, 30]. In a related line of research, human feedback has been used to train agents in simulated environments [10, 24]. There is also a rich literature on using RL to optimize automatic metrics for NLP tasks, such as ROUGE for summarization [50, 65, 45, 15, 19], BLEU for translation [50, 66, 1, 43], and other domains [61, 27, 26]. Finally, there has been extensive research on modifying architectures [22, 59] and pre-training procedures [70, 36, 49, 60, 53, 14] for improving summarization performance.

## 3    Method and experiment details

### 3.1    High-level methodology

Our approach is similar to the one outlined in [73], adapted to the batch setting. We start with an initial policy that is fine-tuned via supervised learning on the desired dataset (in our case, the Reddit TL;DR summarization dataset). The process (illustrated in Figure 2) then consists of three steps that can be repeated iteratively.

**Step 1:  Collect samples from existing policies and send comparisons to humans.** For each Reddit post, we sample summaries from several sources including the current policy, initial policy, original reference summaries and various baselines. We send a batch of pairs of summaries to our human evaluators, who are tasked with selecting the best summary of a given Reddit post.

**Step 2: Learn a reward model from human comparisons.** Given a post and a candidate summary, we train a reward model to predict the log odds that this summary is the better one, as judged by our labelers.

**Step 3: Optimize a policy against the reward model.** We treat the logit output of the reward model as a reward that we optimize using reinforcement learning, specifically with the PPO algorithm [58].

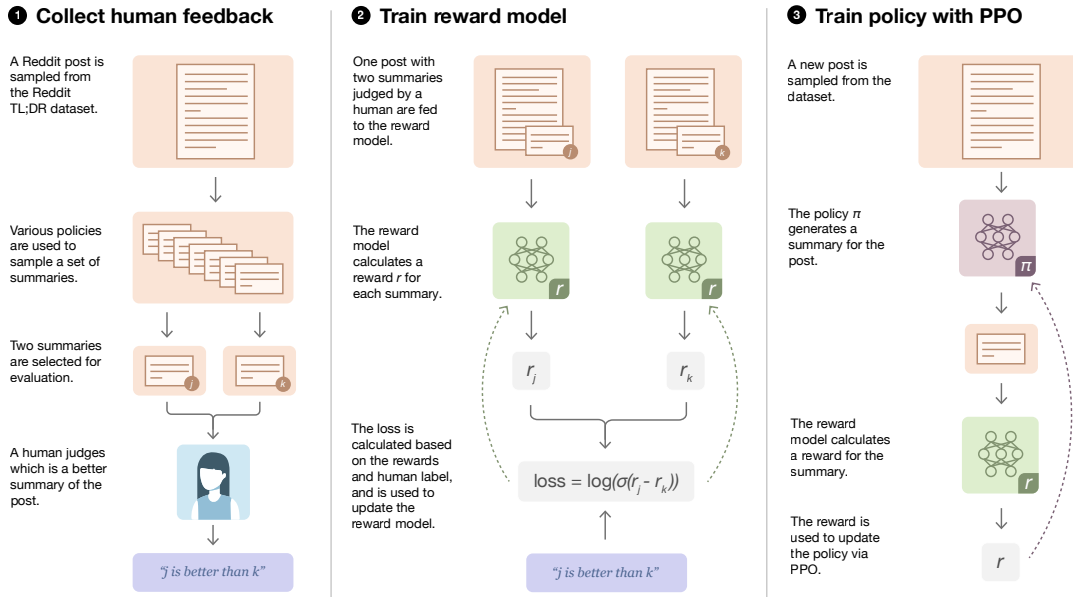

**❶ Collect human feedback**

A Reddit post is sampled from the Reddit TL;DR dataset.

Various policies are used to sample a set of summaries.

Two summaries are selected for evaluation.

A human judges which is a better summary of the post.

*"j is better than k"*

**❷ Train reward model**

One post with two summaries judged by a human are fed to the reward model.

The reward model calculates a reward $r$ for each summary.

The loss is calculated based on the rewards and human label, and is used to update the reward model.

$$\text{loss} = \log(\sigma(r_j - r_k))$$

*"j is better than k"*

**❸ Train policy with PPO**

A new post is sampled from the dataset.

The policy $\pi$ generates a summary for the post.

The reward model calculates a reward for the summary.

The reward is used to update the policy via PPO.

Figure 2: Diagram of our human feedback, reward model training, and policy training procedure.

We provide a more thorough description of our procedure, including details of the reward model and policy training and our quality control process, in the following sections. In practice, rather than precisely iterating this sequence of three steps, we updated our data collection and training procedures over the course of the project while accumulating labels (see Appendix C.6 for details).

## 3.2 Datasets and task

**Datasets.** We use the TL;DR summarization dataset [63], which contains ~3 million posts from `reddit.com` across a variety of topics (subreddits), as well summaries of the posts written by the original poster (TL;DRs). We additionally filter this dataset (see Appendix A) to ensure quality, including using a whitelist of subreddits that are understandable to the general population. Crucially, we also filter to include only posts where the human-written summaries contain between 24 and 48 tokens, to minimize the potential effect of summary length on quality (see Section 4.1 and Appendix F). Our final filtered dataset contains 123,169 posts, and we hold out ~5% as a validation set. For the remainder of this paper, we refer to this dataset simply as TL;DR.

We chose the TL;DR dataset over the more commonly used CNN/DM dataset primarily because very strong performance can be attained on CNN/DM with simple extractive baselines. We find in Section 4.2 that our labelers prefer lead-3 over the CNN/DM reference summaries,[5] and that the supervised T5 model [49] with low-temperature sampling already surpasses the reference summary quality, while copying extensively from the article. On the other hand, simple extractive baselines perform poorly on TL;DR in our human evaluations (see Appendix G.2). Instead of training on CNN/DM, we study the transfer performance of our human feedback models to CNN/DM after being trained to summarize Reddit posts.

**Task.** We define our ground-truth task as producing a model that generates summaries fewer than 48 tokens long that are as good as possible, according to our judgments. We judge summary quality by how faithfully the summary conveys the original post to a reader who can only read the summary and not the post (see Appendix C.5 for further discussion of criteria). Since we have limited capacity to do comparisons, we hire labelers to do the comparisons for us. We rely on detailed procedures to ensure high agreement between labelers and us on the task, which we describe in the next section.

| [r/dating_advice] **First date ever, going to the beach. Would like some tips** |
| --- |
| Hey Reddit! I (20M) would like some tips, because I have my first ever date tomorrow (although I've had a gf for 3 years, but no actual dating happened), and we're going to the beach. |
| I met this girl, we have mutual friends, at a festival a few days ago. We didn't kiss, but we talked, held hands, danced a bit. I asked her to go on a date with me, which was super hard as it is the first time I've asked this to anybody. What I mean to say is, it's not like a standard *first* date because we already spent some time together. |
| I'm really nervous and excited. I'm going to pick her up tomorrow, we're cycling to the beach which will take 30 minutes, and then what? I'm a bit scared. Should I bring something (the weather, although no rain and sunny, is not super so no swimming), should we do something. I'd like all the tips I can get. Thanks! |

| **Human written reference TL;DR** | **6.7B supervised model** | **6.7B human feedback model** |
| --- | --- | --- |
| First date after 3 years in a relationship, going to the beach, terrified. What to bring with me, what to do? | Going on a date with a girl I met a few days ago, going to the beach. What should I bring, what should we do? | Going on my first ever date tomorrow, cycling to the beach. Would like some tips on what to do and bring. I'm a bit nervous and excited. Thanks! |

Table 1: Example of post and samples on the TL;DR dataset, chosen to be particularly short. For random samples (along with posts), see Appendix H and our website.

### 3.3 Collecting human feedback

Previous work on fine-tuning language models from human feedback [73] reported "a mismatch between the notion of quality we wanted our model to learn, and what the humans labelers actually evaluated", leading to model-generated summaries that were high-quality according to the labelers, but fairly low-quality according to the researchers.

Compared to [73], we implement two changes to improve human data quality. First, we transition entirely to the offline setting, where we alternate between sending large batches of comparison data[6] to our human labelers and re-training our models on the cumulative collected data. Second, we maintain a hands-on relationship with labelers:[7] we on-board them with detailed instructions, answer their questions in a shared chat room, and provide regular feedback on their performance. We train all labelers to ensure high agreement with our judgments, and continuously monitor labeler-researcher agreement over the course of the project. See Appendix C.1 and C.5 for details.

As a result of our procedure, we obtained high labeler-researcher agreement: on a subset of comparison tasks, labelers agree with researchers $77\% \pm 2\%$ of the time, while researchers agree with each other $73\% \pm 4\%$ of the time. We provide more analysis of our human data quality in Appendix C.2.

### 3.4 Models

All of our models are Transformer decoders [62] in the style of GPT-3 [47, 4]. We conduct our human feedback experiments on models with 1.3 billion (1.3B) and 6.7 billion (6.7B) parameters.

**Pretrained models.** Similarly to [12, 47], we start with models pretrained to autoregressively predict the next token in a large text corpus. As in [48, 4], we use these models as 'zero-shot' baselines by padding the context with examples of high-quality summaries from the dataset. We provide details on pretraining in Appendix B, and on our zero-shot procedure in Appendix B.2.

**Supervised baselines.** We next fine-tune these models via supervised learning to predict summaries from our filtered TL;DR dataset (see Appendix B for details). We use these supervised models to sample initial summaries for collecting comparisons, to initialize our policy and reward models, and as baselines for evaluation. In our final human evaluations, we use T=0 to sample from all models, as we found it performed better than higher temperatures or nucleus sampling (see Appendix B.1).

To validate that our supervised models are indeed strong baselines for comparison, we run our supervised fine-tuning procedure with our 6.7B model on the CNN/DM dataset, and find that we achieve slightly better ROUGE scores than SOTA models [71] from mid-2019 (see Appendix G.4).

**Reward models.** To train our reward models, we start from a supervised baseline, as described above, then add a randomly initialized linear head that outputs a scalar value. We train this model to predict which summary $y \in \{y_0, y_1\}$ is better as judged by a human, given a post $x$. If the summary preferred by the human is $y_i$, we can write the RM loss as:

$$\text{loss}(r_\theta) = E_{(x, y_0, y_1, i) \sim D}[\log(\sigma(r_\theta(x, y_i) - r_\theta(x, y_{1-i})))]$$

where $r_\theta(x, y)$ is the scalar output of the reward model for post $x$ and summary $y$ with parameters $\theta$, and $D$ is the dataset of human judgments. At the end of training, we normalize the reward model outputs such that the reference summaries from our dataset achieve a mean score of 0.

**Human feedback policies.** We want to use the reward model trained above to train a policy that generates higher-quality outputs as judged by humans. We primarily do this using reinforcement learning, by treating the output of the reward model as a reward for the entire summary that we maximize with the PPO algorithm [58], where each time step is a BPE token.[8] We initialize our policy to be the model fine-tuned on Reddit TL;DR. Importantly, we include a term in the reward that penalizes the KL divergence between the learned RL policy $\pi_\phi^{\text{RL}}$ with parameters $\phi$ and this original supervised model $\pi^{\text{SFT}}$, as previously done in [25]. The full reward $R$ can be written as:

$$R(x, y) = r_\theta(x, y) - \beta \log[\pi_\phi^{\text{RL}}(y|x)/\pi^{\text{SFT}}(y|x)]$$

This KL term serves two purposes. First, it acts as an entropy bonus, encouraging the policy to explore and deterring it from collapsing to a single mode. Second, it ensures the policy doesn't learn to produce outputs that are too different from those that the reward model has seen during training.

For the PPO value function, we use a Transformer with completely separate parameters from the policy. This prevents updates to the value function from partially destroying the pretrained policy early in training (see ablation in Appendix G.1). We initialize the value function to the parameters of the reward model. In our experiments, the reward model, policy, and value function are the same size.

## 4 Results

### 4.1 Summarizing Reddit posts from human feedback

**Policies trained with human feedback are preferred to much larger supervised policies.** Our main results evaluating our human feedback policies on TL;DR are shown in Figure 1. We measure policy quality as the percentage of summaries generated by that policy that humans prefer over the reference summaries in the dataset. Our policies trained with human feedback significantly outperform our supervised baselines on this metric, with our 1.3B human feedback model significantly outperforming a supervised model $10\times$ its size (61% versus 43% raw preference score against reference summaries). Our 6.7B model in turn significantly outperforms our 1.3B model, suggesting that training with human feedback also benefits from scale. Additionally, both of our human feedback models are judged by humans to be superior to the human demonstrations used in the dataset.

**Controlling for summary length.** When judging summary quality, summary length is a confounding factor. The target length of a summary is implicitly part of the summarization task; depending on the desired trade-off between conciseness and coverage, a shorter or longer summary might be better. Since our models learned to generate longer summaries, length could account for much of our quality improvements. We find that after controlling for length (Appendix F), the preference of our human feedback models vs. reference summaries drops by ~5%; even so, our 6.7B model summaries are still preferred to the reference summaries ~65% of the time.

**How do our policies improve over the baselines?** To better understand the quality of our models' summaries compared to the reference summaries and those of our supervised baselines, we conduct an additional analysis where human labelers assess summary quality across four dimensions (or "axes") using a 7-point Likert scale [38]. Labelers rated summaries for coverage (how much important information from the original post is covered), accuracy (to what degree the statements in the summary are stated in the post), coherence (how easy the summary is to read on its own), and overall quality.

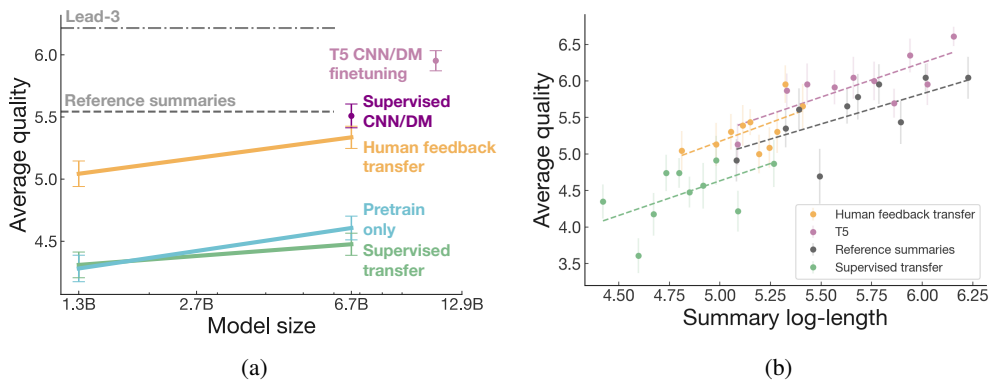

|           | (a) | (b) |
|-----------|-----|-----|

Figure 4: Transfer results on CNN/DM. (a) Overall summary quality on CNN/DM as a function of model size. Full results across axes shown in Appendix G.2. (b) Overall scores vs. length for the 6.7B TL;DR supervised baseline, the 6.7B TL;DR human feedback model, and T5 fine-tuned on CNN/DM summaries. At similar summary lengths, our 6.7B TL;DR human feedback model nearly matches T5 despite never being trained to summarize news articles.

The results (Figure 3) indicate that our human feedback models outperform the supervised baselines across every dimension of quality, but particularly coverage. Although our human labelers had a high bar for giving perfect overall scores, summaries from our 6.7B PPO model achieve a 7/7 overall score 45% of the time (compared to 20% and 23% for the 6.7B supervised baseline and reference summaries, respectively).

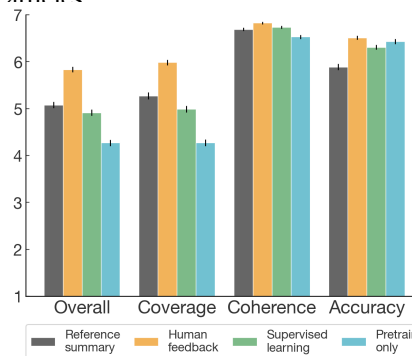

Figure 3: Evaluations of four axes of summary quality on the TL;DR dataset.

## 4.2 Transfer to summarizing news articles

Our human feedback models can also generate excellent summaries of CNN/DM news articles without any further training (Figure 4). Our human feedback models significantly outperform models trained via supervised learning on TL;DR and models trained only on pretraining corpora. In fact, our 6.7B human feedback model performs almost as well as a 6.7B model that was fine-tuned on the CNN/DM reference summaries, despite generating much shorter summaries.

Since our human feedback models transferred to CNN/DM have little overlap in summary length distribution with models trained on CNN/DM, with about half as many tokens on average, they are difficult to compare directly. Thus our evaluations in Figure 4 use a 7-point Likert scale on four quality dimensions, as in Section 4.1 (see Appendix C.5 for labeler instructions). In Figure 4b we show the average overall score at different summary lengths, which suggests our human feedback models would perform even better if they generated longer summaries. Qualitatively, CNN/DM summaries from our human feedback models are consistently fluent and reasonable representations of the article; we show examples on our website and in Appendix H.

## 4.3 Understanding the reward model

**What happens as we optimize the reward model?** Optimizing against our reward model is supposed to make our policy align with human preferences. But the reward model isn't a perfect representation of our labeler preferences, as it has limited capacity and only sees a small amount of comparison data from a relatively narrow distribution of summaries. While we can hope our reward model generalizes to summaries unseen during training, it's unclear how much one can optimize against the reward model until it starts giving useless evaluations.

To answer this question, we created a range of policies optimized against an earlier version of our reward model, with varying degrees of optimization strength, and asked labelers to compare samples from them to the reference summaries. Figure 5 shows the results for PPO at a range of KL penalty

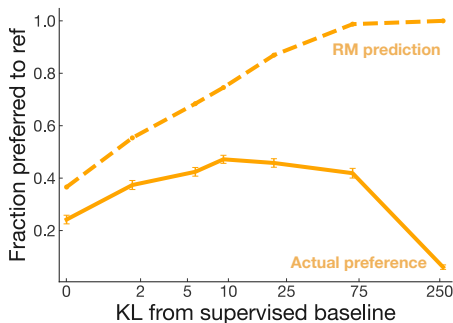

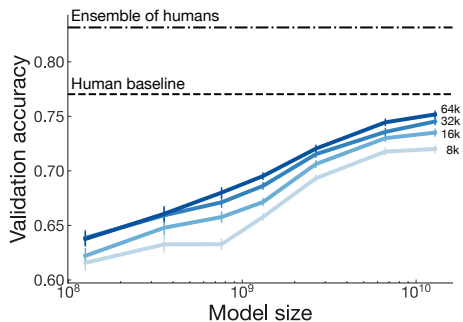

Figure 5: Preference scores versus degree of reward model optimization. Optimizing against the reward model initially improves summaries, but eventually overfits, giving worse summaries. This figure uses an earlier version of our reward model (see rm3 in Appendix C.6). See Appendix H.2 for samples from the KL 250 model.

Figure 6: Reward model performance versus data size and model size. Doubling amount of training data leads to a ~1.1% increase in reward model validation accuracy, whereas doubling the model size leads to a ~1.8% increase. The 6.7B model trained on all data begins approaching the accuracy of a single human.

coefficients ($\beta$). Under light optimization, the models improve (according to labelers). However, as we optimize further, true preferences fall off compared to the prediction, and eventually the reward model becomes anti-correlated with human preferences. Though this is clearly undesirable, we note that this over-optimization also happens with ROUGE (see [45] and Appendix G.3). Similar behavior has been observed in learned reward functions in the robotics domain [5].

**How does reward modeling scale with increasing model and data size?**    We conduct an ablation to determine how data quantity and model size affect reward modeling performance. We train 7 reward models ranging from 160M to 13B parameters, on 8k to 64k human comparisons from our dataset. We find that doubling the training data amount leads to a ~1.1% increase in the reward model validation set accuracy, whereas doubling the model size leads to a ~1.8% increase (Figure 6).

**What has the reward model learned?**    We probe our reward model by evaluating it on several validation sets. We show the full results in Appendix G.6, and highlight them here. We find that our reward models generalize to evaluating CNN/DM summaries (Appendix G.7), agreeing with labeler preferences 62.4% and 66.5% of the time (for our 1.3B and 6.7B models, respectively). Our 6.7B reward model nearly matches the inter-labeler agreement value of 66.9%.

We also find that our reward models are sensitive to small but semantically important details in the summary. We construct an additional validation set by having labelers make minimal edits to summaries to improve them. Our RMs prefer the edited summaries almost as often (79.4% for 1.3B and 82.8% for 6.7B) as a separate set of human evaluators (84.1%). Further, when comparing the reference summaries to perturbed summaries where the participants' roles are reversed, our models reliably select the original summary (92.9% of the time for 1.3B, 97.2% for 6.7B). However, our RMs are biased towards longer summaries: our 6.7B RM prefers improving edits that make the summary shorter only 62.6% of the time (vs. 76.4% for humans).

### 4.4   Analyzing automatic metrics for summarization

**Evaluation.**    We study how well various automatic metrics act as predictors for human preferences, and compare them to our RMs. Specifically, we examine ROUGE, summary length, amount of copying from the post,[9] and log probability under our baseline supervised models. We present a full matrix of agreement rates between these metrics in Appendix G.7.

We find that our learned reward models consistently outperform other metrics, even on the CNN/DM dataset on which it was never trained. We also find that ROUGE fails to track sample quality as our

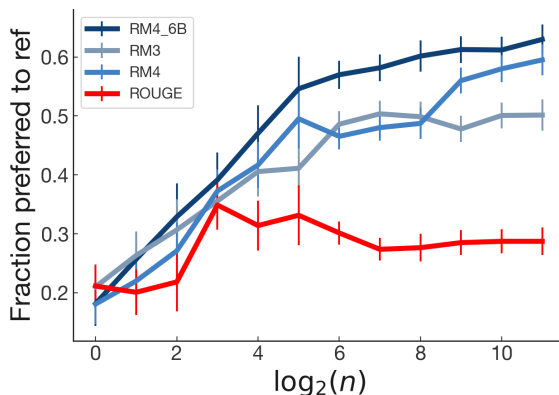

Figure 7: Summary quality as a function of metric optimized and amount of optimization, using best-of-N rejection sampling. We evaluate ROUGE, our main reward models, and an earlier iteration of the 1.3B model trained on approximately 75% as much data (see Table 11 for details). ROUGE appears to peak both sooner and at a substantially lower preference rate than all reward models. Details in Appendix G.3.

models improve. While ROUGE has ~57% agreement with labelers when comparing samples from our supervised baseline models, this drops to ~50% for samples from our human feedback model.

Similarly, log probability agreement with humans drops to $\leq 50\%$ on comparisons between samples from our human feedback models, while our RMs still perform above chance (62%). Scaling up the size of the supervised model does not reliably improve log probability's agreement with labelers.

**Optimization.** In Figure 7, we show that optimizing ROUGE using a simple optimization scheme doesn't consistently increase quality, as has been noted in [45]. Optimization against ROUGE peaks both sooner and at a substantially lower quality rate than optimization against our reward models.

## 5 Discussion

**Limitations.** One limitation of our work is the time and cost required to produce our final models. Notably, fine-tuning our 6.7B model with RL required approximately 320 GPU-days. Our data collection procedure is also expensive compared to prior work — the training set took thousands of labeler hours and required significant researcher time to ensure quality. For this reason, we were unable to collect baselines such as an equivalent amount of high-quality human demonstrations for supervised baselines. See D for more discussion. We leave this ablation to future work. Nevertheless, we believe reward modeling is more likely to scale to tasks where it is extremely skill-intensive or time-consuming to provide good demonstrations.

**Future directions.** The methods in this paper could be applied to any task where humans can compare samples, including dialogue, machine translation, question answering, speech synthesis, and music generation. We expect this method to be particularly important for generating long samples, where the distributional shift and degeneracy of maximum likelihood samples can be problematic. It may be possible to improve sample efficiency by training to predict feedback across many tasks [42].

We are particularly interested in scaling human feedback to tasks where humans can't easily evaluate the quality of model outputs. In this setting, it is particularly challenging to identify whether an ML system is aligned with the human designer's intentions. One approach is to train ML systems to help humans perform the evaluation task quickly and accurately [9].

There is also a rich landscape of human feedback methods beyond binary comparisons that could be explored for training models [28, 17, 44, 64]. For example, we could solicit high-quality demonstrations from labelers, have labelers edit model outputs to make them better, or have labelers provide explanations for why they preferred one model output over another. All of this feedback could be leveraged as a signal to train more capable reward models and policies.

**Broader impacts.**   The techniques we explore in this paper are generic techniques that could be used in a wide variety of machine learning applications, for any task where it is feasible for humans to evaluate the quality of model outputs. Thus, the potential implications are quite broad.

Our research is primarily motivated by the potential positive effects of aligning machine learning algorithms with the designer's preferences. Many machine learning applications optimize simple metrics which are only rough proxies for what the designer intends. This can lead to problems, such as Youtube recommendations promoting click-bait [11]. In the short term, improving techniques for learning from and optimizing human preferences directly may enable these applications to be more aligned with human well-being.

In the long term, as machine learning systems become more capable it will likely become increasingly difficult to ensure that they are behaving safely: the mistakes they make might be more difficult to spot, and the consequences will be more severe. For instance, writing an inaccurate summary of a news article is both easy to notice (one simply has to read the original article) and has fairly low consequences. On the other hand, imitating human driving may be substantially less safe than driving to optimize human preferences. We believe that the techniques we explore in this paper are promising steps towards mitigating the risks from such capable systems, and better aligning them with what humans care about.

Unfortunately, our techniques also enable malicious actors to more easily train models that cause societal harm. For instance, one could use human feedback to fine-tune a language model to be more persuasive and manipulate humans' beliefs, or to induce dependence of humans on the technology, or to generate large amounts of toxic or hurtful content intended to harm specific individuals. Avoiding these outcomes is a significant challenge for which there are few obvious solutions.

Large-scale models trained with human feedback could have significant impacts on many groups. Thus, it is important to be careful about how we define the 'good' model behavior that human labelers will reinforce. Deciding what makes a good summary is fairly straightforward, but doing this for tasks with more complex objectives, where different humans might disagree on the correct model behavior, will require significant care. In these cases, it is likely not appropriate to use researcher labels as the 'gold standard'; rather, individuals from groups impacted by the technology should be included in the process to define 'good' behavior, and hired as labelers to reinforce this behavior in the model.

We chose to train on the Reddit TL;DR dataset because the summarization task is significantly more challenging than on CNN/DM. However, since the dataset consists of user-submitted posts with minimal moderation, they often contain content that is offensive or reflects harmful social biases. This means our models can generate biased or offensive summaries, as they have been trained to summarize such content. For this reason, we recommend that the potential harms of our models be thoroughly studied before deploying them in user-facing applications.

Finally, by improving the ability of machine learning algorithms to perform tasks that were previously only achievable by humans, we are increasing the likelihood of many jobs being automated, potentially leading to significant job loss. Without suitable policies targeted at mitigating the effects of large-scale unemployment, this could also lead to significant societal harm.

**Acknowledgements**

We'd like to thank Beth Barnes for help with labeler hiring and general encouragement; Geoffrey Irving for guidance on earlier iterations of the project and inspiring conversations; Ben Mann, Tom Brown, Nick Ryder, and Melanie Subbiah for training and evaluating our pretrained models; Chris Hesse, Eric Sigler, Benjamin Chess, Christopher Berner, Clemens Winter, Mateusz Litwin, and many others for supporting us through computing infrastructure improvements and maintenance; Scott Gray for writing fast GPU kernels; Arvind Neelakantan and Wojciech Kryscinski for discussions on how to present the work, experiment design, and what datasets to use; Shan Carter for help designing the main diagram; Douwe Kiela, Zach Lipton, and Alex Irpan for providing feedback on the paper; and Gretchen Krueger for co-writing the model card accompanying the paper.

Finally, we'd like to thank all of our contractors for providing the data that was essential for training the models in this paper, including: Emill Jayson Caypuno, Rachelle Froyalde, Cyra Denura, Alex Malek, Isik Agil, Reshmi Patel, William Yap, Natalie Silver, Erol Akbaba, Jennifer Brillo, Alexandra

Uifalean, Morris Stuttard, Russell Bernandez, Tasmai Dave, Rachel Wallace, Jenny Fletcher, Jian Ouyang, Justin Dill, Maria Orzek, Megan Niffenegger, William Sells, Emily Mariner, Andrew Seely, Lychelle Ignacio, Jelena Ostojic, Nhan Tran, Purev Batdelgar, Valentina Kezic, Michelle Wilkerson, Kelly Guerrero, Heather Scott, Sarah Mulligan, Gabriel Ricafrente, Kara Bell, Gabriel Perez, and Alfred Lee.

## Footnotes

[4]Throughout the paper, error bars represent 1 standard error.

[5]We manually check this result in Appendix E and find we generally agree with labeler ratings.

[6]Our decision to collect comparisons rather than Likert scores is supported by recent work, e.g. [37].

[7]We recruited labelers from a freelancing platform, Upwork, and two labeling services, Scale and Lionbridge.

[8]Note that the reward model only gives rewards for entire summaries, and not at intermediate time steps. In RL terminology, each episode terminates when the policy outputs the EOS token, and the discount factor $\gamma = 1$.

[9]We measure copying by computing the longest common subsequence of bigrams with the original Reddit post or news article, and dividing by the number of bigrams in the summary.

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
