[Supplementary Material]

# Appendix

## Table of Contents

# A  TL;DR dataset details

Here, we discuss the pre-processing steps that we apply to the TL;DR dataset. We first remove all duplicate posts by checking the text body, finding that there are nearly 20,000 exact duplicates. We then re-parse the TL;DR carefully using a set of heuristics, and filter to use only top-level posts (rather than comments). We also filter out any post that is from a subreddit not in our 'subreddit whitelist' (see Table 2 for the distribution over subreddits), any post where the title starts with some variant of 'Edit' or 'Update',[10] and posts that contain certain topics (such as graphic sex or suicide) using heuristics. Finally, to ensure the posts are short enough to fit into the context length of our models, we filter out any post whose body is longer than 512 tokens. This resulted in a set of 287,790 posts filtered by body but not summary, of which we hold out approximately 5% as a validation set. We used this set of posts for RL training since our RL procedure does not require reference summaries.

We next perform additional filtering on the parsed reference summaries that we use for training our supervised baselines. Specifically, we remove summaries where the TL;DR starts with variants of 'Edit', 'Update', or 'P.S.', we heuristically remove summaries with certain levels of profanity, and we remove summaries that are less than 24 tokens or more than 48 tokens. As discussed in Section 4.1, since our RL models tend to generate summaries on the upper end of the allowed length limit, this length filtering ensures that there is enough length overlap between the RL summaries and reference summaries for us to perform a length-controlled analysis. Additionally, we found that summaries shorter than 16 tokens were usually of low quality. We later verified that the summaries we filtered out were lower

| Subreddit | # posts | % of dataset |
|---|---|---|
| relationships | 63324 | 54.25% |
| AskReddit | 15440 | 13.23% |
| relationship_advice | 8691 | 7.45% |
| tifu | 7685 | 6.58% |
| dating_advice | 2849 | 2.44% |
| personalfinance | 2312 | 1.98% |
| Advice | 2088 | 1.79% |
| legaladvice | 1997 | 1.71% |
| offmychest | 1582 | 1.36% |
| loseit | 1452 | 1.24% |
| jobs | 1084 | 0.93% |
| self | 1048 | 0.90% |
| BreakUps | 838 | 0.72% |
| askwomenadvice | 688 | 0.59% |
| dogs | 638 | 0.55% |
| running | 567 | 0.49% |
| pettyrevenge | 548 | 0.47% |
| needadvice | 528 | 0.45% |
| travel | 452 | 0.39% |
| Parenting | 435 | 0.37% |
| weddingplanning | 433 | 0.37% |
| Pets | 366 | 0.31% |
| Dogtraining | 362 | 0.31% |
| cats | 324 | 0.28% |
| AskDocs | 283 | 0.24% |
| college | 264 | 0.23% |
| GetMotivated | 169 | 0.14% |
| books | 161 | 0.14% |
| Cooking | 114 | 0.10% |

Table 2: Number of posts in the training set of our filtered Reddit TL;DR dataset by subreddit.

quality according to our reward model — more than 0.5 nats worse on average (i.e. they are predicted to be $\exp(0.5) \approx 1.6$ times less likely to be preferred). Our final TL;DR dataset contains 123,169 posts including summaries, again with about 5% held out as a validation set. We use 1913 of these validation articles for model selection during development; the evaluations in this paper exclude these articles.

Note that, from Table 2 we can see that about two thirds of our TL;DR dataset consists of posts relating to relationships or relationship advice, which is a fairly specific domain. This raises potential concerns about the generality of our models, though their strong transfer performance on CNN/DM news articles suggests they are not unreasonably specialized to relationship advice.

| Model size | n_layers | d_model | n_heads | Max LR | Max batch size |
|---|---|---|---|---|---|
| 1.3B | 24 | 2048 | 16 | 2e-4 | 512 |
| 3B | 32 | 2560 | 32 | 1.6e-4 | 512 |
| 6.7B | 32 | 4096 | 32 | 1.2e-4 | 512 |
| 13B | 40 | 5120 | 40 | 1e-4 | 1024 |

Table 3: Hyperparameters for our models of various sizes.

Figure 8: The sweep we conducted for determining our sampling procedure, varying the temperature and the 'top p' value for nucleus sampling. While we didn't do a large enough test to determine whether nucleus sampling is better or worse than moderate-temperature sampling, we found that very low temperature sampling is better than both on this task.

## B    Further model training details

### B.1    Hyperparameters

All models follow the standard Transformer architecture, with 2048 learned position embeddings. All models are trained with fp16 activations and the Adam optimizer [31]. Nearly all supervised baselines, reward models, and reinforcement learning models are trained with fp32 weights; the exception is our TL;DR supervised baselines, which were trained with fp16 weights.[11] All models are trained with the same byte-pair encoding as in [48].

During pretraining, the models were trained to predict the next token on a large text corpus consisting of Commoncrawl, Webtext [48], books, and Wikipedia. Training lasts between 1-3 epochs on each, for a total of 200-300 billion tokens. Learning rate follows a cosine schedule, with a short warmup, decaying to 10% of the maximum value. The batch size ramped up throughout training to some maximum, with each input having 2048 tokens. Hyperparameters for each model are shown in Table 3.

For supervised baselines, we initialize models from the pretrained models. We decay the learning rate with a cosine schedule, using an initial learning rate chosen from a log linear sweep of at least 7 values. This resulted in learning rates of 6.35e-5, 5.66e-5, 2.83e-5, and 2.83e-5 for our TL;DR models of size 1.3B, 3B, 6.7B, and 13B respectively, and a learning rate of 2.38e-5 for our CNN/DM 6.7B model. We use a batch size of 128, and run for a single epoch.

For reward modeling, we initialize to the supervised baseline, but with a reward head on top with weights initialized according to $\mathcal{N}(0, 1/(d_{model} + 1))$ [20]. We train for one epoch, decaying the

| Trained models | Format | Max tokens |
|---|---|---|
| TL;DR (supervised, RL) | SUBREDDIT: r/{subreddit}<br>TITLE: {title}<br>POST: {post}<br>TL;DR: | 512 |
| Transfer from TL;DR to CNN/DM (supervised, RL) | {article}<br>TL;DR: | 512 |
| TL;DR (pretrained) | {context_stuffed_with_examples}<br>=====<br>Subreddit: r/{subreddit}<br>Title: {title}<br>{post}<br>TL;DR: | 1999 |
| CNN/DM (supervised) | Article: {article}<br>TL;DR: | 1999 |
| CNN/DM (pretrained) | {context_stuffed_with_examples}<br>=====<br>Article: {article}<br>TL;DR: | 1999 |

Table 4: Formats used for the context for each of our trained models on the TL;DR and CNN/DM datasets.

learning rate with a cosine schedule, using an initial learning rate chosen from a log linear sweep of at least 7 values. We also sweep over between 3 and 10 seeds, and choose the reward model that performs best on the development portion of the validation set, as we find that both the data iteration order and reward head initialization affect results [13]. For our main results, the 1.3B and 6.7B reward models had learning rates of 1.5e-5 and 5e-6, respectively. We use a batch size of 64, and run for a single epoch.

For PPO, we run with separate policy and value networks, initializing our policies to the supervised baseline, and our value functions to the reward model. We set $\gamma = 1$ and $\lambda = 0.95$ for the advantage estimation [57] and do 4 epochs of optimization for each batch of rollouts. We used a linear learning rate decay schedule, with initial learning rates of 1.5e-5 for the 1.3B model and 7e-6 for the 6.7B model, based on small amounts of experimentation and rough model size extrapolation. We used a KL coefficient of 0.05 for both of the main runs we report results for (except when we explicitly vary this value in the reward model optimization graphs). We use a batch size of 512 for the 1.3B model and 256 for the 6.7B model, and run for 1 million episodes.

## B.2 Input format

Our model always receives a byte-pair encoded string of a fixed size. When the input is too small, we pad from the beginning of the input with a padding token, and if the input is too long we truncate the post/article field at newlines to stay under the limit.

When sampling from models pretrained only on our pretrain mixture and not fine-tuned on TL;DR, we follow [48] and instead of padding with a padding token, we pad the beginning of the context with examples of posts/articles and high-quality summaries. We use as many examples as will fit in the token limit, with the examples formatted the same way as the main input. Table 4 documents the formats we used (with pythonic format strings).

# C  Human data collection details

## C.1  Process for ensuring high-quality human data

We first detail the procedures we use to ensure high-quality data. While these procedures became more rigorous over the course of the project, they generally involved four steps.

**Step 0: Understanding the task ourselves.** To understand the task, we first do many summary comparisons ourselves. We also hire a small number of human labelers[12] to do comparisons, and discuss our disagreements. We then draft instructions for a larger set of human labelers.

**Step 1: Labeler onboarding.** Labelers are hired from Upwork, a freelancing platform, as well as two labeling services, Scale and Lionbridge. Labelers first complete a (paid) training process where they label summaries on a shared set of data. For some comparisons, labelers get immediate feedback about which summary was chosen by us, and why, to help them calibrate. We retain labelers that pass a minimum threshold for speed and agreement with us. To allow for a customizable labeler interface, we built our own website for data collection (see Appendix C.4).

**Step 2: Collecting comparison data.** Next, we have labelers evaluate a large batch of comparisons on our website, which generates the bulk of our data. Before comparing two summaries directly, we have labelers write their 'naive interpretations' of summaries without seeing the original post. We've found this helpful for evaluating summaries, as they surface points of ambiguity in the summary that might not have been detected if the summary was read after the original post. After doing naive interpretations, labelers do comparisons by assigning a value on a 9-point scale for how confident they are that summary A is better than summary B (or the converse).

**Step 3: Providing labeler feedback.** After collecting the comparison data, we can look at agreement rates between labelers. While most comparisons are only given to a single labeler, each labeler gets about 10-20% questions from a shared pool for calibration purposes. We can both attempt to use these statistics as crude measures of quality, and show cases of disagreements to workers to help them improve their labels.

**Step 4: Researcher comparison calibrations.** We occasionally also do the task ourselves, to measure agreement rates between each labeler and us. This is used for quality assessment (see C.2). We also calculate per-labeler "high confidence" thresholds, by finding the confidence value on the Likert scale for each labeler such that we expect labels above this threshold to agree with us 80% of the time on average. For the purposes of reward model selection, we filter the validation set to contain only these higher confidence labels. For the entire process, we keep a high communication bandwidth with labelers: we use a shared chat room for labelers to ask clarifying questions and discuss difficult comparisons amongst themselves, host office hours, and occasionally have one-on-one video calls with labelers to discuss points of disagreement.

We keep good labelers throughout the lifetime of the project, while firing the lowest-performing workers.

## C.2  Assessing human feedback quality

We assess labeler accuracy by comparing the labeler's preferred summary with the summary we prefer (ignoring the confidence level). We exclude comparisons where either the labeler or researcher expresses indifference. This gives us an agreement rate, in theory ranging from 0% (perfect disagreement) to 100% (perfect agreement). For our 2-way comparisons, a random labeler would get 50% agreement.

To obtain our main number comparing labeler-researcher to researcher-researcher agreement, we restrict ourselves to comparisons between summaries from our 1.3B supervised baseline, because this subset of the data has the most researcher-labeled data. On this subset, labelers agree with researchers $77\% \pm 2\%$ of the time, while researchers agree with each other $73\% \pm 4\%$ of the time. We believe substantial noise comes from comparisons being quite difficult and subjective.

In general, agreement rates range from about 65% for the least proficient labelers and most difficult comparisons (comparing two high-temperature samples from a single RL policy) to about 85% for

Figure 9: (a) The website we made to collect data from labelers. (b) Naive interpretations of summaries on the website.

the most proficient labelers and easiest comparisons (comparing a high-temperature sample from a supervised baseline to the reference summary). Averaging over all workers, weighted by their volume, gives us an estimated agreement rate of $73\% \pm 3\%$ for our reward model training corpus.

Labelers agree with each other 72% of the time in the training corpus. This suggests we could get more reliable labels by aggregating labels from multiple workers on the same comparison. Indeed, on the subset of the training data for which we have enough shared comparisons, taking the modal label from 3 labelers increases their agreement rate with researchers from 72% to 77%. However, we usually collect only one label per comparison, in order to maximize label throughput.

On the evaluations for Figure 1, labelers agreed with researchers $73\% \pm 3\%$ of the time, and labelers agreed with each other $73\% \pm 2\%$ of the time.

Agreement rate between researchers ranged from about 65% on the most difficult comparisons (comparing two high-temperature samples from a single RL policy), to about 80% on the easiest comparisons (comparing a high-temperature sample from a supervised baseline to the human reference summary), to about 95% in cases where we discussed the comparisons with each other.

Overall we believe that quality is fairly high. Our attempts to filter data generally hurt reward model accuracy. For example, using the confidence thresholds mentioned above, we found that while lower-confidence labels were less useful than high-confidence labels for improving reward model accuracy, they were still better to include than to omit. Similarly, leaving out workers with poorer agreement rates did not help.

## C.3  Labeler demographics

When training machine learning models with human feedback, the humans providing the feedback are essential in reinforcing the desired model behavior. If we are to scale human feedback to train models on more complex tasks, where humans might disagree about what the desired model behavior should be, it's important for members of groups that will be impacted by the model to be included in the labeler population.

To provide more transparency into our labeler demographics, we provide results from a survey given to our labelers in Table 5. The survey was optional, anonymous, and it was made clear that the results would not affect hiring or firing decisions. We find that our labelers span a range of ethnicities, nationalities, ages, and genders, and educational backgrounds, but are more likely to be White and American.

## C.4  Labeler website

Since we hired and trained our own set of labelers, rather than using a crowdsourcing website such as Amazon Mechanical Turk, we built our own website to allow for a standardized, customizable user interface for all labelers. Each labeler created a separate profile, allowing us to assign different sets of comparisons to different labelers. The website contains different renderers for different kinds

| What gender do you identify as? | |
|---|---|
| Male | 38.1% |
| Female | 61.9% |
| Nonbinary / other | 0% |
| **What ethnicities do you identify as?** | |
| White / Caucasian | 42.9% |
| Southeast Asian | 23.8% |
| Indigenous / Native American / Alaskan Native | 9.6% |
| East Asian | 4.8% |
| Middle Eastern | 4.8% |
| Latinx | 4.8% |
| My ethnic identity isn't listed | 9.6% |
| **What is your nationality?** | |
| American | 45% |
| Filipino | 30% |
| South African | 5% |
| Serbian | 5% |
| British | 5% |
| Turkish | 5% |
| Indian | 5% |
| **What is your age?** | |
| 20-29 | 42.9% |
| 30-39 | 23.8% |
| 40-49 | 23.8% |
| 50-59 | 9.5% |
| 60+ | 0% |
| **What is your highest attained level of education?** | |
| Less than high school degree | 0% |
| High school degree | 14.3% |
| Undergraduate degree | 57.1% |
| Master's degree | 23.3% |
| Doctorate degree | 4.8% |

Table 5: Demographic data from 21 of our labelers who participated in our voluntary survey.

of questions, including naive interpretations, summary comparisons, and Likert evaluations along different axes, along with room for labelers to express concerns with the question or explanations for their decision. Screenshots from the website are shown in Figure 9. Data collected from the website can be easily ported into a central database containing all of our human data.

## C.5 Instructions for labelers

Here we provide more detail on the specific instructions given to labelers for comparing summaries, and for doing Likert evaluations of summaries along axes of quality. We produced separate sets of instructions for evaluating Reddit posts, and for evaluating CNN/DM news articles. For Reddit instructions, we first describe Reddit in general and provide a table that translates Reddit-specific lingo into common parlance.

**Instructions for comparing summaries.** We show an excerpt of the instructions given to labelers for making comparisons in Table 6. In addition to these instructions, we provide an example labeled comparison between Reddit summaries, and also example naive interpretations for summaries.

**Instructions for evaluating summaries along axes of quality.** We provide a separate set of detailed instructions for labelers for the 7-point Likert evaluations. We first introduce each of the 4 axes of quality we consider, giving an overview of coherence, accuracy, coverage, and overall score (shown in Table 7). We also provide a brief rubric for giving scores of 1, 4, and 7, along with several Reddit summaries annotated with our own judgments of quality along each of these axes (with explanations).

What makes for a good summary? Roughly speaking, a good summary is a shorter piece of text that has the essence of the original – tries to accomplish the same purpose and conveys the same information as the original post. We would like you to consider these different dimensions of summaries:

**Essence:** is the summary a good representation of the post?

**Clarity:** is the summary reader-friendly? Does it express ideas clearly?

**Accuracy:** does the summary contain the same information as the longer post?

**Purpose:** does the summary serve the same purpose as the original post?

**Concise:** is the summary short and to-the-point?

**Style:** is the summary written in the same style as the original post?

Generally speaking, we give higher weight to the dimensions at the top of the list. Things are complicated though – none of these dimensions are simple yes/no matters, and there aren't hard and fast rules for trading off different dimensions. This is something you'll pick up through practice and feedback on our website.

Table 6: An excerpt from the instructions we gave to labelers for doing comparisons.

Finally, we provide a FAQ section that answers common questions raised by the small initial set of labelers we assigned to this task.

For CNN/DM, we provide the same set of instructions, except we add some additional clarifications for how to judge news articles. We specifically ask labelers to place less emphasis on fluidity of sentences (because the reference summaries were originally written in bullet-point form, and we didn't want labelers to penalize this), and to place less emphasis on the summary matching the intent of the article (which was important for Reddit summaries).

In terms of quality control, we conducted a smaller version of the quality control process described in Appendix C.1: we first labeled a small set of summaries ourselves along each axis to understand points of confusion, then we wrote the instructions document to provide to labelers, then we had a small number of labelers do a trial of the task to catch any remaining bugs or points of confusion, and finally we onboarded a larger set of labelers onto the task while remaining available to answer any questions.

### C.6   Composition of the labeled dataset

Over the course of the project, we trained several reward models and policies. Each batch of summaries that we sent to the labelers were sampled from a variety of policies. We didn't have a systematic plan for which policies to sample from; rather, we chose what seemed best at the time in the spirit of exploratory research. Every time we trained a reward model, we trained on all labels we had collected so far. Successive models also benefited from improved hyperparameters and dataset cleaning. Our results could likely be replicated with a simpler, more systematic approach.

In general, as we hire new labelers and as existing labelers perform the task more, it is possible that there is 'labeler drift', where the set of criteria used by labelers to evaluate summaries gradually shifts over time. This could lead to a regression in labeler-researcher disagreement, or lead to some policies becoming more or less preferred over time. To help guard against this, in most batches we include comparisons between samples from our supervised baseline and reference summaries, and measure the frequency with which the workers prefer one over the other. If this number drifts over time, it's an indication that our workers' preferences are also changing. However, we generally found that this preference number stayed relatively constant, within noise.

Table 8 lists the policies we trained by supervised finetuning on the TL;DR dataset, as well as the reward models, trained on successively larger datasets of human labels. Table 9 lists the RL policies.

## Coherence

For this axis, answer the question "how coherent is the summary on its own?" A summary is coherent if, when read by itself, it's easy to understand and free of English errors. A summary is not coherent if it's difficult to understand what the summary is trying to say. Generally, it's more important that the summary is understandable than it being free of grammar errors.

**Rubric:**

Score of 1: The summary is impossible to understand.
Score of 4: The summary has mistakes or confusing phrasing that make it a bit hard to understand.
Score of 7: The summary is perfectly clear.

## Accuracy

For this axis, answer the question "does the factual information in the summary accurately match the post?" A summary is accurate if it doesn't say things that aren't in the article, it doesn't mix up people, and generally is not misleading. If the summary says anything at all that is not mentioned in the post or contradicts something in the post, it should be given a maximum score of 5. (If you are confused about how to use '6', see the FAQ!)

**Rubric:**

Score of 1: The summary is completely wrong, made up, or exactly contradicts what is written in the post.
Score of 4: The summary says at least one substantial thing that is not mentioned in the post, or that contradicts something in the post.
(Score of 5: The summary says anything, no matter how small, that is not mentioned in the post, or that contradicts something in the post.)
Score of 7: The summary has no incorrect statements or misleading implications.

## Coverage

For this axis, answer the question "how well does the summary cover the important information in the post?" A summary has good coverage if it mentions the main information from the post that's important to understand the situation described in the post. A summary has poor coverage if someone reading only the summary would be missing several important pieces of information about the situation in the post. A summary with good coverage should also match the purpose of the original post (e.g. to ask for advice).

**Rubric:**

Score of 1: The summary contains no information relevant to the post.
Score of 4: The summary is missing at least 1 important piece of information required to understand the situation.
Score of 7: The summary covers all of the important information required to understand the situation.

## Overall quality

For this axis, answer the question "how good is the summary overall at representing the post?" This can encompass all of the above axes of quality, as well as others you feel are important. If it's hard to find ways to make the summary better, give the summary a high score. If there are lots of different ways the summary can be made better, give the summary a low score.

**Rubric:**

Score of 1: The summary is terrible.
Score of 4: The summary is an okay representation of the post, but could be significantly improved.
Score of 7: The summary is an excellent representation of the post.

Table 7: Instructions given to labelers for evaluating summaries along four different axes of quality.

| Supervised policy name | # Parameters |
|---|---|
| sup1 | 750M |
| sup2 | 1.3B |
| sup3 | 1.3B |
| sup3_6b | 6.7B |
| sup4 | 1.3B |
| sup4_6b | 6.7B |

| Reward model name | # Parameters |
|---|---|
| rm1 | 1.3B |
| rm2 | 6.7B |
| rm3 | 1.3B |
| rm3_6b | 6.7B |
| rm4 | 1.3B |
| rm4_6b | 6.7B |

Table 8: Left: supervised baselines. sup4 and sup4_6b are the final supervised baselines used throughout the paper. Right: reward models. rm4 and rm4_6b are the final reward models used throughout the paper.

| RL policy name | # Parameters | Objective | Initialization | KL coefficient | KL(ppo, sup) |
|---|---|---|---|---|---|
| sup3 ppo rm1 | 1.3B | rm1 | sup3 | 0.35 | 1.8 |
| sup4 ppo rm3 1 | 1.3B | rm3 | sup4 | 0.10 | 3.8 |
| sup4 ppo rm3 2 | 1.3B | rm3 | sup4 | 0.07 | 9.4 |
| sup4 ppo rm3 3 | 1.3B | rm3 | sup4 | 0.05 | 19.0 |
| sup4 ppo rm4 | 1.3B | rm4 | sup4 | 0.05 | 18.0 |
| sup4_6b ppo rm4_6b | 6.7B | rm4_6b | sup4_6b | 0.05 | 14.0 |

Table 9: PPO policies. sup4 ppo rm4 and sup4_6b ppo rm4_6b are the final policies used throughout the paper.

| BoN policy name | Objective | Base policy | N | KL(BoN, sup) |
|---|---|---|---|---|
| sup2 bo8 rm1 | rm1 | sup2 | 8 | 1.2 |
| sup3 bo8 rm1 | rm2 | sup3 | 8 | 1.2 |
| sup3 bo63 rm2 | rm2 | sup3 | 63 | 3.2 |
| sup4 bo8 rm3 | rm3 | sup4 | 8 | 1.2 |
| sup4 bo64 rm3 | rm3 | sup4 | 64 | 3.2 |
| sup4 bo128 rm3 | rm3 | sup4 | 128 | 3.9 |
| sup4 bo256 rm3 | rm3 | sup4 | 256 | 4.5 |
| sup4 bo512 rm3 | rm3 | sup4 | 512 | 5.2 |
| sup4 bo128 rm3_6b | rm3_6b | sup4 | 128 | 3.9 |
| sup4 bo256 rm3_6b | rm3_6b | sup4 | 256 | 4.5 |

Table 10: Best-of-N policies. KL divergence is computed analytically as KL(boN, sup) = log N - (N-1)/N.

We also explored a simple alternative to reinforcement learning: Sample N summaries from a supervised baseline at temperature 0.7, score them with a reward model, and take the summary with the highest score. This best-of-N (BoN) procedure is effectively a mildly optimized policy requiring no training. These policies are named in Table 10, and samples from them form part of the training data.

Table 11 lists the source policies for the training data for each reward model.

| Reward model | Policy0 | Policy1 | Label count |
|---|---|---|---|
| rm1 | ref | sup1 | 5404 |
| | sup1 | sup1 | 5386 |
| rm2 | ref | sup1 | 5404 |
| | | sup2 | 12779 |
| | | sup2 bo8 rm1 | 1426 |
| | | sup3_6b | 1424 |
| | sup1 | sup1 | 5386 |
| | | Continued on next page | |

| Reward model | Policy0 | Policy1 | Label count |
|---|---|---|---|
| | sup2 | sup2 | 11346 |
| | | sup2 bo8 rm1 | 1376 |
| | | sup3_6b | 1383 |
| | sup2 bo8 rm1 | sup3_6b | 1390 |
| rm3, rm3_6b | ref | sup1 | 5404 |
| | | sup2 | 12779 |
| | | sup2 bo8 rm1 | 1426 |
| | | sup3 | 438 |
| | | sup3 bo63 rm2 | 447 |
| | | sup3 bo8 rm2 | 887 |
| | | sup3 ppo rm1 | 884 |
| | | sup3_6b | 1424 |
| | sup1 | sup1 | 5386 |
| | sup2 | sup2 | 11346 |
| | | sup2 bo8 rm1 | 1376 |
| | | sup3_6b | 1383 |
| | sup2 bo8 rm1 | sup3_6b | 1390 |
| | sup3 | sup3 bo8 rm2 | 428 |
| | | sup3 ppo rm1 | 416 |
| | sup3 bo63 rm2 | sup3 bo8 rm2 | 432 |
| | | sup3 ppo rm1 | 444 |
| | sup3 bo8 rm2 | sup3 ppo rm1 | 855 |
| rm4, rm4_6b | ref | sup1 | 5404 |
| | | sup2 | 12779 |
| | | sup2 bo8 rm1 | 1426 |
| | | sup3 | 438 |
| | | sup3 bo63 rm2 | 447 |
| | | sup3 bo8 rm2 | 887 |
| | | sup3 ppo rm1 | 884 |
| | | sup3_6b | 1424 |
| | | sup4 | 1335 |
| | | sup4 bo128 rm3 | 602 |
| | | sup4 bo128 rm3_6b | 203 |
| | | sup4 bo256 rm3 | 307 |
| | | sup4 bo256 rm3_6b | 101 |
| | | sup4 bo512 rm3 | 52 |
| | | sup4 bo64 rm3 | 52 |
| | | sup4 bo8 rm3 | 393 |
| | | sup4 ppo rm3 1 | 981 |
| | | sup4 ppo rm3 2 | 215 |
| | | sup4 ppo rm3 3 | 208 |
| | | sup4_6b | 104 |
| | sup1 | sup1 | 5386 |
| | sup2 | sup2 | 11346 |
| | | sup2 bo8 rm1 | 1376 |
| | | sup3_6b | 1383 |
| | sup2 bo8 rm1 | sup3_6b | 1390 |
| | sup3 | sup3 bo8 rm2 | 428 |
| | | sup3 ppo rm1 | 416 |
| | sup3 bo63 rm2 | sup3 bo8 rm2 | 432 |
| | | sup3 ppo rm1 | 444 |
| | sup3 bo8 rm2 | sup3 ppo rm1 | 855 |
| | sup4 | sup4 | 1051 |
| | | sup4 ppo rm3 1 | 395 |

|               |                    |                    | Label count |
| Reward model  | Policy0            | Policy1            |             |
| ------------- | ------------------ | ------------------ | ----------- |
|               | sup4 bo128 rm3     | sup4 bo128 rm3     | 288         |
|               |                    | sup4 bo256 rm3     | 582         |
|               | sup4 bo128 rm3_6b  | sup4 bo128 rm3_6b  | 95          |
|               |                    | sup4 bo256 rm3_6b  | 203         |
|               | sup4 bo512 rm3     | sup4 ppo rm3 3     | 216         |
|               |                    | sup4_6b            | 60          |
|               | sup4 bo64 rm3      | sup4 ppo rm3 2     | 218         |
|               |                    | sup4_6b            | 55          |
|               | sup4 bo8 rm3       | sup4 ppo rm3 1     | 752         |
|               | sup4 ppo rm3 1     | sup4 ppo rm3 1     | 372         |
|               | sup4 ppo rm3 2     | sup4 ppo rm3 2     | 4256        |
|               |                    | sup4_6b            | 215         |
|               | sup4 ppo rm3 3     | sup4 ppo rm3 3     | 4037        |
|               |                    | sup4_6b            | 216         |

Table 11: Training data for reward models. "ref" refers to human reference summaries.

## C.7 Example comparison tasks

To give a sense of the difficulty of the comparisons task, we provide example comparisons between two summaries generated by our 6.7B human feedback model. In Table 12 we show both a random comparison drawn from the TL;DR dataset, and a cherry-picked comparison (selected from 10 comparisons where labelers disagreed) to illustrate the trade-off between accuracy in coverage that can occur when labelers conduct evaluations.

**Random TL;DR comparison**

**POST**

Subreddit: r/Pets

TITLE: What do you use for flea control?

My family has tried literally EVERYTHING to control the fleas in our neighborhood (Frontline, Advantage, Diatomaceous Earth, Dawn Dishsoap, etc!) and nothing has worked. I have spoken to lots of pet owners in my area (I work as a vet assistant) and many are reporting similar results, where fleas are becoming resistant to the usually recommended treatments. The only thing that has worked so far is Comfortis, but I've read of several dogs having reactions to it that can be pretty severe. My dogs are fine, we've used it for about a year now, but I don't like the idea of harming them or putting them at risk.

Giving them baths with blue Dawn dish soap does kill all the fleas, but it does nothing to prevent more from coming back, obviously. It only kills on contact, and we are NOT going to over bath them because that isn't healthy either. We're looking for something that lasts.

Does anyone else have experience with this, or any detailed information on Comfortis and if it does serious damage to your pet's system? Yes, I know I am a vet assistant. My boss strictly recommends Frontline and literally will not listen to me when I tell him it doesn't work and my dogs are still covered in fleas and we have to use Comfortis because it is the only thing that gives them relief. He is not a resource in this case.

Just wanted to see what other pet owners (specifically ones in San Diego) do for fleas...the ones we have here are mutants or something, because almost nothing works on them!

**Summary A:** Fleas are developing resistance to most flea control products (including Comfortis). Looking for something that lasts long term that doesn't harm my dogs. Does anyone have experience with any of the listed products?

**Summary B:** Nothing has worked on our fleas, we are looking for something that lasts, Comfortis is not a long term solution. Does anyone else have experience with flea control or have information on Comfortis?

**Hard TL;DR comparison**

**POST**

Subreddit: r/weddingplanning

TITLE: Feeling major anxiety about dress shopping.

So, not really sure if I'm asking for advice or just a small rant. We got engaged March 2, 2013. From day 1 we've been struggling through the planning. At first, it was arguing with his parents about us getting married in a church. And then it was an argument about which venue to have the reception. We finally have the venue booked and the church matter settled. Now that's out of the way, I suddenly have this pit in my stomach

My mom left me when I was 14. I've basically done everything on my own and I have really been ok about it. I'm sure it's not of the norm for me to feel so disassociated about the whole thing, but I am. I'm suppose to go look at wedding dresses this Friday. I am feeling super anxious because I don't know if trying on wedding dresses is going to turn me into a blubbering baby about not having a mom.

My future mother-in-law is suppose to come with me to help look. I worry about turning into that blubbering baby and offending her. I don't want her thinking that I don't appreciate her being there.

Aside from me worrying about becoming a giant baby, I've also been having issues with my bridal party. While I haven't made any official choices, I have ideas of who I want involved. That would be my best friend, my sister, and my future sister-in-law. My first choice for a MOH is my best friend. However, she lives out of state, and is in a medical program for school. So her visit time is severely limited. My sister feels entitled to be the MOH, despite the fact that we are not close at all. So getting people together to get any kind of wedding stuff done is almost impossible.

**Summary A:** I'm having doubts about whether or not to try on wedding dresses. I am also having doubts about my bridal party's ability to get things done.

**Summary B:** I think I'm going to turn into a blubbering baby and offend my mother-in-law.

Table 12: Top: Example of a random comparison task on the TL;DR dataset between two summaries from our 6.7B human feedback model. Comparison chosen randomly from the validation set. Bottom: An example of a difficult comparison task on the TL;DR dataset. Chosen by looking at comparisons between supervised baseline summaries with at least 4 labeler judgements and with at least 40% vote for each summary. Cherry-picked out of 10 to highlight an accuracy-coverage tradeoff. Summary A is inaccurate since the author does not explicitly say she is having doubts about trying on wedding dresses. Summary B is entirely accurate but does not capture the general essence of the post. In this case, 4 workers chose A and 3 workers chose B. For more comparisons, see our website.

# D   Choice of baselines

In testing our human feedback techniques, we collected a large amount of high-quality data from human labelers. In order to compare fairly against supervision-based techniques, we would have needed to spend a similar amount of labeler time collecting high quality demonstrations, and used those to fine-tune a model via supervised learning. Because this is prohibitively expensive, we do not provide such a baseline.

Existing prior work such as PEGASUS [70] has studied supervised methods on a dataset very similar to ours (the /r/tifu subset of TL;DR). However, they use much smaller (500M parameters) models, and report that their model outputs are worse than the human reference summaries, according to human evaluations. Thus, due to our limited labeler budget for evaluation, we decided to use our own supervised and zero-shot models as baselines (after sanity-checking the ROUGE performance of our supervised models), as well as T5 [49].

T5 models [49] are pretrained and fine-tuned in a similar way to our supervised baselines, but they use an encoder-decoder architecture. We used T5 outputs which were obtained via beam search decoding, as described in [49]. We also carefully account for differences in tokenization between model outputs.[13]

# E   CNN/DM lead-3 vs reference summaries

On the CNN/DM dataset, our labelers significantly preferred lead-3 (a summary consisting of the first 3 sentences of the article) to reference summaries. In part this is due to longer summaries receiving higher coverage scores and lead-3 being 50% longer, as shown in Table 13.

| Policy | Length (stdev) | Quality | Quality increase / 100 char. |
|---|---|---|---|
| ref | 314 (119) | 5.54 | 0.14 |
| lead-3 | 475 (114) | 6.23 | 0.34 |

Table 13: How length affects overall quality on CNN/DM for lead-3 and reference summaries.

However, if we use a linear regression (similar to the procedure in Appendix F) to predict what lead-3 performance would be if its average length were reduced to 314 characters, we still find a quality of 5.68, modestly higher than the reference summaries. Moreover, for lead-3 to even achieve parity with the reference summaries seems to call into question the need for abstractive summarization or sophisticated ML methods, since a simple extractive baseline can match a perfect imitation of the reference summaries.

We wanted to understand labeler behavior on these comparisons, to ensure that it was not an error. To do this, we examined a sample of our labeler's judgments ourselves. We found that in 20/143 cases labelers preferred lead-3 by 3 points or more, and that excluding these datapoints would raise the relative score of the reference summaries by about 0.5 points.[14] We were surprised to see the reference summaries performing so poorly in a significant fraction of cases, so we looked at labeler's explanations and confirmed they made sense.

We found that two features of the reference summaries explained most of its underperformance. First, 13 of these 20 summaries omitted one of the key points from the article—the highlights are often written for a reader who had already seen the title of the article, even though the titles are not included in the CNN/DM dataset. Second, 10 of these 20 summaries actually introduced new information not present in the original article. From the perspective of labelers this information is totally confabulated and so led to lower scores. A likely explanation for these errors is that the reference summaries are extracted from "highlights" on the news sites rather than being a straightforward summary of the article. These failures are common enough that they significantly impact the average quality of the reference summaries, and the effects seem to be large relative to quality differences between ML models.

Overall we believe that labeler judgments were reasonable in these cases, and that it is potentially problematic to treat the "highlights" in the CNN/DM dataset as reference summaries. You can view all of our labeler's judgments on CNN/DM at our website.

Figure 10: (a) A length-controlled version of Figure 1, using the procedure described in Appendix F. Controlling for length reduces the relative preference of our human feedback models, however they are still preferred to the reference summaries. (b) Plotting model quality for different summary lengths on the TL;DR dataset. Our 6.7B human feedback model outperforms both the 6.7B supervised baseline and the reference summaries (horizontal line at 0.5) across lengths.

# F    Controlling for summary length

As discussed in Section 4.1, the length of a summary is a confounding factor for evaluating summary quality; depending on the desired trade-off between conciseness and coverage, a shorter or longer summary might be better. Our models generate summaries that are longer than the reference summaries, as this led to higher labeler preference given the 24-48 token limit for our task. Here we describe the procedure we use to attempt to control for length.

To calculate a single length-controlled preference number, we train a logistic regression model to predict the human-preferred summary on our dataset of human comparisons. We provide this model with 2 features: the identity of each policy, and the log ratio of the summary lengths. To calculate the length-controlled preference value between two policies, we simply give each policy ID to our trained logistic regression model and set the log length ratio to zero (see Figure 10a). In Figure 10b we examine summary quality across a range of summary lengths on TL;DR. We find that our human feedback model outperforms the supervised baseline across all length values.

For CNN/DM, we use a similar procedure as described above to control for length, except using a linear regression model to predict the Likert rating from 1-7. We show the expected quality increase for making summaries 100 characters longer in Table 14, which suggests our human feedback models would perform better if they generated longer summaries.

| Policy | Length (stdev) | Quality (1-7) | Quality ↗ / 100 char. |
|---|---|---|---|
| sl(tldr)-1.3b | 138 (34) | 4.26 | 0.68 |
| sl(tldr)-6.7b | 127 (31) | 4.41 | 0.38 |
| gpt-1.3b | 141 (41) | 4.11 | 0.63 |
| gpt-6.7b | 142 (36) | 4.6 | 0.3 |
| rl(tldr)-1.3b | 166 (30) | 4.86 | 1.28 |
| rl(tldr)-6.7b | 175 (30) | 5.25 | 0.87 |
| sl(cnn)-6.7b | 300 (103) | 5.4 | 0.37 |
| ref | 314 (119) | 5.54 | 0.14 |
| lead-3 | 475 (114) | 6.23 | 0.34 |
| T5 | 316 (95) | 5.92 | 0.3 |

Table 14: How length affects overall quality on CNN/DM. We show average length and quality scores for various policies, and how much the summary quality increases on average per 100 added characters.

# G  Additional results

## G.1  Value function ablation

In this section, we conduct an ablation comparing using separate parameters for the value function and policy, against using a shared network as done in [73]. The results, shown in Figure 11, clearly indicate that using separate networks outperforms the latter. On the other hand, having separate networks increases the memory requirements of running RL fine-tuning. Having separate networks also allows us to initialize the value function to be the learned reward model that is being optimized.

Figure 11: Comparing the reward obtained by optimizing with separate value function and reward model parameters to shared parameters.

## G.2  Evaluating policies along axes of quality

We show the full results of the evaluations of policies on a 7-point Likert scale along different axes of quality; for TL;DR this is shown in Figure 12, and for CNN/DM this is shown in Figure 13. It is evident that on both datasets coverage correlates strongly with overall score across models, and all models achieve a high coherence score.

## G.3  Studying best-of-N optimization

A natural way to evaluate an automatic evaluation metric is to see the extent to which optimizing against it leads to high performance according to humans. One way to assess this is to use best-of-N as an (inefficient) optimization technique — this has the benefits of being simple and invariant to monotonic transformations. We report results for up to best-of-2048 on ROUGE and three of our reward models in Figure 7, using samples from the 1.3B supervised baseline. The results suggest that optimizing against ROUGE significantly under-performs optimizing against our reward models. The data also suggests ROUGE degrades with too much optimization much faster than our reward models.

With increasing N, the best-of-N policies get higher average reward. Similarly, by decreasing the KL coefficient $\beta$, the PPO policies get higher average reward. We found that at a given average reward, the best-of-N and PPO policies have similar quality as judged by human labelers (not shown). However, the PPO policy is farther from the supervised baseline than best-of-N is, as measured by the KL divergence.[15]

## G.4  ROUGE scores

In Figure 14a and 14b, we show the ROUGE scores of our models on the TL;DR and CNN/DM datasets, respectively. We report results with T=0, consistent with our human evaluations. We found that temperature has an (often significant) impact on ROUGE score, and we did a thorough sweep to verify that the best temperature setting is T=0.

Figure 12: Evaluating TL;DR policies on a 7-point Likert scale along several axes of quality.

| Model | ROUGE-1 | ROUGE-2 | ROUGE-L |
|---|---|---|---|
| ProphetNet [67] | 44.20 | 21.17 | 40.69 |
| T5 [49] | 43.52 | 21.55 | 40.69 |
| **Our 6.7B supervised model** | 42.49 | 19.84 | 39.53 |
| CNN-2sent-hieco-RBM [71] | 42.04 | 19.77 | 39.42 |

Table 15: Comparing the ROUGE score of our 6.7B supervised model on CNN/DM to recent SOTA models from the literature. Without any summarization-specific engineering, our model achieves ROUGE scores better than SOTA models from mid-2019, indicating that it is a strong baseline for comparison.

Figure 13: Evaluating CNN/DM policies on a 7-point Likert scale along several axes of quality.

On TL;DR, we find that our human feedback models obtain a slightly lower ROUGE score than the supervised models at $T = 0$, further indicating that ROUGE correlates poorly with human preferences. For supervised models, lowering temperature has a larger impact than increasing model size. Interestingly, at higher temperatures, our feedback models actually outperform supervised counterparts (not shown).

On CNN/DM, ROUGE agrees with our human evaluations that our human feedback models transfer better than our supervised models. However, unsurprisingly, supervised CNN/DM models still achieve much higher ROUGE. In Table 15, we show the ROUGE results on CNN/DM for our 6.7B supervised baseline and various models from the literature. We find that our model achieves ROUGE scores less than T5 [49], but slightly greater than the CNN-2sent-hieco-RBM model from [71], which was SOTA for abstractive summarization on CNN/DM in mid-2019 according to the NLP-progress leaderboard.[16]

### G.5 Bigram overlap statistics

In Table 16, we show the bigram overlap statistics for our models on the TL;DR and CNN/DM datasets as a proxy for how much the summaries copy frmo the post. As in Section 4.4, we compute the longest common subsequence of bigrams with the original Reddit post or news article, and dividing by the number of bigrams in the summary. We find that models evaluated on CNN/DM

Figure 14: ROUGE scores for our models on (a) the TL;DR dataset, and (b) the CNN/DM dataset.

| Evaluated on TL;DR | | |
|---|---|---|
| **Model** | **Model size** | **Bigram overlap %** |
| GPT | 1.3B | 66.7% |
| GPT | 3B | 72.7% |
| GPT | 6.7B | 61.4% |
| GPT | 13B | 75.9% |
| Supervised (TL;DR) | 1.3B | 49.0% |
| Supervised (TL;DR) | 3B | 48.7% |
| Supervised (TL;DR) | 6.7B | 48.9% |
| Supervised (TL;DR) | 13B | 48.0% |
| Human feedback (TL;DR) | 1.3B | 53.3% |
| Human feedback (TL;DR) | 6.7B | 46.0% |
| **Evaluated on CNN/DM** | | |
| **Model** | **Model size** | **Bigram overlap %** |
| GPT | 1.3B | 76.3% |
| GPT | 6.7B | 76.2% |
| Supervised (TL;DR) | 1.3B | 59.5% |
| Supervised (TL;DR) | 6.7B | 56.9% |
| Human feedback (TL;DR) | 1.3B | 64.8% |
| Human feedback (TL;DR) | 6.7B | 51.2% |
| Supervised (CNN/DM) | 1.3B | 66.0% |
| T5 | 11B | 68.8% |
| reference | — | 36.8% |

Table 16: Bigram overlap statistics on the TL;DR dataset (top) and the CNN/DM dataset (bottom). Models trained on CNN/DM copy significantly more than models trained on TL;DR.

(whether or not they were trained on CNN/DM) generally copy more than models evaluated on TL;DR. Further, our supervised and human feedback models copy less than our pretrained models.

### G.6 Reward model validation sets

In this section, we report results evaluating our reward models on various manually constructed validation sets, shown in Tables 17 and 18. Notably, we asked our humans to produce a small dataset of edits, by having them make improvements to existing summaries (either reference summaries or supervised baseline summaries). Our 6.7B reward model prefer the improved summaries at a similar rate to humans (who do not know which summary has been edited).

Our reward models are also sensitive to sentence shuffling (whereas metrics like ROUGE are largely not), and are able to detect when the roles portrayed in the summary have been switched. On the other hand, our reward models sometimes exhibit preference for poor artificial summaries, such as

| RM size | Edit length | RM prefers edit | Human prefers edit | RM, human agree |
|---|---|---|---|---|
| | Shorter | 63.6% | 76.2% | 62.1% |
| 1.3B | Longer | 86.8% | 88.6% | 79.6% |
| | Avg. | 81.2% | 85.6% | 75.4% |
| | Shorter | 66.0% | 76.2% | 65.5% |
| 6.7B | Longer | 89.2% | 88.6% | 80.2% |
| | Avg. | 83.7% | 85.6% | 76.7% |

Table 17: Comparing reward model and human preference of summaries that were edited by humans to make them better. For each summary, the human labeler that makes the comparison is different than the labeler that wrote the edit. The agreement numbers do not include comparisons where the labeler's preference was marked as 'uncertain'.

| | | Preference % of Summary A | |
|---|---|---|---|
| Summary A | Summary B | 1.3B RM | 6.7B RM |
| Original summary | Reversed roles | 93.1% | 97.4% |
| lead-3 | Shuffled lead-3 | 68.1% | 75.5% |
| rand-3 | Shuffled rand-3 | 60.8% | 76.1% |
| Post title | Random title | 97.4% | 98.5% |
| Post title | Random title from same subreddit | 98.8% | 97.2% |
| Post title | Post title repeated twice | 84.6% | 58.4% |
| (r/tifu only) Reference summary | Ref + "What should I do?" | 34.3 % | 74.5% |
| Reference summary | lead-3 | 63.0% | 56.4% |
| Reference summary | lead-2 | 71.0% | 73.8% |
| Reference summary | rand-3 | 69.5% | 59.5% |

Table 18: Reward model performance on various manually constructed validation sets. In all cases, Summary A is intended to be better than Summary B, and thus a higher preference % is generally better. 'rand-3' indicates a baseline where 3 random sentences are taken from the post; however these sentences are kept in the order in which they appear in the post. 'Original summary' is either the reference summary or a summary from our supervised baselines. r/tifu is a subreddit whose purpose is sharing embarrassing stories (not asking for advice).

the post title copied twice, or asking for advice at the end of the summary. In Table 19, we show examples where our model is sensitive to small, semantically meaningful changes in the summary.

### G.7 Measuring agreement between different evaluation metrics

We are interested in understanding the relationship between different metrics for evaluating summaries. To do this, we compute agreement between various metrics, including automatic metrics and humans, for different subsets of the data for which we have human evaluations. To remove policy quality as a confounding variable, all of the summary comparisons are generated by the same policy at the same temperature value. In Table 20, we use samples from our 1.3B supervised model at T=0.7 on TL;DR; Table 21 has comparisons from our 6.7B supervised model at T=0.7 on TL;DR; Table 22 has comparisons from our 6.7B human feedback model at T=0.7 on TL;DR; and Table 23 has comparisons from our 6.7B supervised baseline trained on CNN/DM.

Our 6.7B reward model generally agrees with labelers as much as other labelers, although an ensemble of labelers does better. On the other hand, ROUGE generally has poor agreement, as does log probability under the supervised baselines, with simple heuristics like copying (longest common subsequence of bigrams with the article) and length often performing comparably.

| Edited summary | Reward Δ |
|---|---|
| Crush on girl I haven't seen in 4 years. She doesn't like me and I ~~don't~~ **still** like her. What do? | +0.64 |
| A girl told me she ~~loved~~ **liked** me, she ended up picking another guy over me, that guy badly influenced her, and now I'm here alone thinking what could've been. | +0.82 |
| I tried to show my friend a picture of my tarantula and she smashed my phone with all her might and now I lost a good ~~friend~~ **phone**. | -0.64 |
| Boyfriend still FB stalks his high school **ex** girlfriend from time to time and told me when he was very drunk that she was his first love. | +0.73 |
| I've become pathetic, pining after ~~a guy~~ **my ex**. Would like to reach state of less pathetic. If more info is necessary, please let me know. | +0.69 |
| I have body issues (body acne/scarring and weight issues) that prevent me from having a normal life without shame and prevent me from having a better **s**ex life with my BF. | +1.0 |
| Do you take someone back after they've turned you ~~down~~ **off**, even if you can't see them in person or are they just not worth the risk? | +0.52 |

Table 19: Qualitative examples showing the change in reward of the reward model on human-generated edits to TL;DR summaries that make the summaries better. Examples are randomly selected from the set where the edit distance was less than 5 and the magnitude of change in reward was greater than 0.5. Text in strike-through was removed from the original summary in the edit, and text in bold was added. The reward model is sensitive to small but semantically meaningful changes in the summary, although it makes errors on occasion.

| TL;DR 1.3B sup. T=0.7 | researcher | labeler | labeler ensemble | length | copying | ROUGE | 1.3B sup logprob | 1.3B RM | 6.7B sup logprob | 6.7B RM |
|---|---|---|---|---|---|---|---|---|---|---|
| researcher | 73.4% ±4.1% | 77.7% ±2.1% | 84.4% ±3.3% | 55.5% ±4.3% | 62.3% ±4.1% | 59.1% ±4.2% | 61.8% ±4.8% | 72.2% ±4.5% | 62.8% ±4.7% | 78.0% ±3.9% |
| labeler | 77.7% ±2.1% | 68.6% ±1.7% | 74.4% ±2.0% | 54.4% ±1.3% | 58.0% ±1.2% | 57.7% ±1.3% | 58.7% ±2.0% | 65.8% ±2.0% | 61.9% ±2.1% | 70.8% ±1.8% |
| labeler ensemble | 84.4% ±3.3% | 74.4% ±2.0% | — | 60.6% ±4.0% | 62.7% ±3.8% | 59.0% ±3.9% | 59.5% ±4.4% | 71.0% ±3.9% | 59.5% ±4.3% | 72.5% ±3.8% |
| length | 55.5% ±4.3% | 54.4% ±1.3% | 60.6% ±4.0% | — | 50.1% ±1.3% | 58.6% ±1.2% | 28.9% ±2.1% | 52.6% ±2.3% | 27.6% ±2.0% | 54.3% ±2.3% |
| copying | 62.3% ±4.1% | 58.0% ±1.2% | 62.7% ±3.8% | 50.1% ±1.3% | — | 51.9% ±1.2% | 61.6% ±2.3% | 57.8% ±2.3% | 60.9% ±2.2% | 55.5% ±2.2% |
| ROUGE | 59.1% ±4.2% | 57.7% ±1.3% | 59.0% ±3.9% | 58.6% ±1.2% | 51.9% ±1.2% | — | 49.5% ±2.3% | 56.4% ±2.2% | 51.1% ±2.3% | 59.2% ±2.3% |
| 1.3B sup. logprob | 61.8% ±4.8% | 58.7% ±2.0% | 59.5% ±4.4% | 28.9% ±2.1% | 61.6% ±2.3% | 49.5% ±2.3% | — | 58.7% ±2.3% | 92.7% ±1.2% | 60.6% ±2.3% |
| 1.3B RM | 72.2% ±4.5% | 65.8% ±2.0% | 71.0% ±3.9% | 52.6% ±2.3% | 57.8% ±2.3% | 56.4% ±2.2% | 58.7% ±2.3% | — | 58.8% ±2.2% | 78.8% ±1.8% |
| 6.7B sup. logprob | 62.8% ±4.7% | 61.9% ±2.1% | 59.5% ±4.3% | 27.6% ±2.0% | 60.9% ±2.2% | 51.1% ±2.3% | 92.7% ±1.2% | 58.8% ±2.2% | — | 61.5% ±2.2% |
| 6.7B RM | 78.0% ±3.9% | 70.8% ±1.8% | 72.5% ±3.8% | 54.3% ±2.3% | 55.5% ±2.2% | 59.2% ±2.3% | 60.6% ±2.3% | 78.8% ±1.8% | 61.5% ±2.2% | — |

Table 20: Agreement rates between humans and various automated metrics on TL;DR 1.3b supervised model at T=0.7. Standard errors estimated via bootstrapping. Note: in the entry for labeler vs. labeler ensemble, the ensembles are slightly smaller than for other comparisons because we need to exclude the labeler being predicted. All ensembles have at least 3 workers.

| TL;DR 6.7B sup. T=0.7 | labeler | labeler ensemble | length | copying | ROUGE | 1.3B sup logprob | 1.3B RM | 6.7B sup logprob | 6.7B RM |
|---|---|---|---|---|---|---|---|---|---|
| labeler | 70.8% ±2.6% | 73.1% ±2.9% | 56.9% ±0.6% | 56.4% ±0.6% | 56.9% ±0.6% | 54.5% ±1.2% | 67.5% ±1.1% | 54.3% ±1.2% | 69.7% ±1.1% |
| labeler ensemble | 73.1% ±2.9% | — | 55.0% ±5.1% | 54.5% ±4.8% | 66.7% ±4.7% | 61.1% ±11.4% | 77.8% ±9.7% | 55.6% ±11.7% | 77.8% ±10.0% |
| length | 56.9% ±0.6% | 55.0% ±5.1% | — | 50.5% ±0.6% | 60.2% ±0.6% | 26.9% ±1.1% | 59.5% ±1.2% | 26.4% ±1.1% | 60.3% ±1.1% |
| copying | 56.4% ±0.6% | 54.5% ±4.8% | 50.5% ±0.6% | — | 54.4% ±0.6% | 59.3% ±1.1% | 57.9% ±1.2% | 60.2% ±1.2% | 58.0% ±1.2% |
| ROUGE | 56.9% ±0.6% | 66.7% ±4.7% | 60.2% ±0.6% | 54.4% ±0.6% | — | 48.7% ±1.2% | 58.1% ±1.2% | 47.7% ±1.2% | 58.4% ±1.2% |
| 1.3B sup. logprob | 54.5% ±1.2% | 61.1% ±11.4% | 26.9% ±1.1% | 59.3% ±1.1% | 48.7% ±1.2% | — | 53.3% ±1.2% | 91.9% ±0.6% | 53.8% ±1.2% |
| 1.3B RM | 67.5% ±1.1% | 77.8% ±9.7% | 59.5% ±1.2% | 57.9% ±1.2% | 58.1% ±1.2% | 53.3% ±1.2% | — | 54.1% ±1.2% | 78.8% ±1.0% |
| 6.7B sup. logprob | 54.3% ±1.2% | 55.6% ±11.7% | 26.4% ±1.1% | 60.2% ±1.2% | 47.7% ±1.2% | 91.9% ±0.6% | 54.1% ±1.2% | — | 54.5% ±1.2% |
| 6.7B RM | 69.7% ±1.1% | 77.8% ±10.0% | 60.3% ±1.1% | 58.0% ±1.2% | 58.4% ±1.2% | 53.8% ±1.2% | 78.8% ±1.0% | 54.5% ±1.2% | — |

Table 21: Agreement rates between humans and various automated metrics on TL;DR 6.7B supervised model at T=0.7. Standard errors estimated via bootstrapping. Note: in the entry for labeler vs. labeler ensemble, the ensembles are slightly smaller than for other comparisons because we need to exclude the labeler being predicted. All ensembles have at least 3 workers.

| TL;DR 6.7B RL T=0.7 | labeler | labeler ensemble | length | copying | ROUGE | 1.3B sup logprob | 1.3B RM | 6.7B sup logprob | 6.7B RM |
|---|---|---|---|---|---|---|---|---|---|
| labeler | 60.4% ±5.9% | 66.0% ±7.6% | 55.8% ±2.2% | 52.7% ±2.1% | 49.9% ±2.1% | 48.0% ±2.2% | 57.4% ±2.0% | 47.3% ±2.2% | 62.3% ±2.1% |
| labeler ensemble | 66.0% ±7.6% | — | 80.0% ±8.9% | 65.0% ±10.6% | 35.0% ±10.5% | 45.0% ±11.1% | 75.0% ±9.8% | 40.0% ±10.5% | 75.0% ±9.8% |
| length | 55.8% ±2.2% | 80.0% ±8.9% | — | 48.1% ±2.2% | 50.3% ±2.1% | 30.0% ±2.1% | 62.0% ±2.1% | 30.4% ±2.0% | 59.8% ±2.2% |
| copying | 52.7% ±2.1% | 65.0% ±10.6% | 48.1% ±2.2% | — | 52.0% ±2.2% | 64.2% ±2.1% | 56.7% ±2.2% | 64.4% ±2.1% | 53.4% ±2.2% |
| ROUGE | 49.9% ±2.1% | 35.0% ±10.5% | 50.3% ±2.2% | 52.0% ±2.2% | — | 50.5% ±2.2% | 52.0% ±2.3% | 51.1% ±2.3% | 54.5% ±2.1% |
| 1.3B sup. logprob | 48.0% ±2.2% | 45.0% ±11.1% | 30.0% ±2.1% | 64.2% ±2.1% | 50.5% ±2.2% | — | 47.0% ±2.2% | 90.2% ±1.3% | 46.1% ±2.2% |
| 1.3B RM | 57.4% ±2.0% | 75.0% ±9.8% | 62.0% ±2.1% | 56.7% ±2.2% | 52.0% ±2.3% | 47.0% ±2.2% | — | 45.7% ±2.1% | 71.4% ±2.0% |
| 6.7B sup. logprob | 47.3% ±2.2% | 40.0% ±10.5% | 30.4% ±2.0% | 64.4% ±2.1% | 51.1% ±2.3% | 90.2% ±1.3% | 45.7% ±2.1% | — | 44.7% ±2.1% |
| 6.7B RM | 62.3% ±2.1% | 75.0% ±9.8% | 59.8% ±2.2% | 53.4% ±2.2% | 54.5% ±2.1% | 46.1% ±2.2% | 71.4% ±2.0% | 44.7% ±2.1% | — |

Table 22: Agreement rates between humans and various automated metrics on TL;DR 6.7B human feedback optimized model at T=0.7. Standard errors estimated via bootstrapping. Note: in the entry for labeler vs. labeler ensemble, the ensembles are slightly smaller than for other comparisons because we need to exclude the labeler being predicted. All ensembles have at least 3 workers.

# H  Samples

## H.1  Random samples

Here we provide non-cherry-picked samples and human evaluations for various models. In Tables 25-26 we show samples on the TL;DR dataset, and in Tables 27-28 we show samples on the CNN/DM dataset (where we truncate the article for brevity). See our website for more uncurated policy samples.

## H.2  Overoptimized samples

We show examples of samples from a policy overoptimized to rm3. The summaries, while clearly long, low quality, and full of idiosyncrasies, do still reflect the rough gist of the post.

| CNN/DM 6.7B sup. T=0.3 | labeler | labeler ensemble | length | copying | ROUGE | 1.3B sup logprob | 1.3B RM | 6.7B sup logprob | 6.7B RM |
|---|---|---|---|---|---|---|---|---|---|
| labeler | 66.9% ±4.3% | 74.5% ±6.8% | 62.4% ±1.4% | 49.6% ±1.4% | 55.2% ±1.4% | 45.7% ±1.4% | 64.8% ±1.4% | 47.6% ±1.4% | 66.5% ±1.3% |
| labeler ensemble | 74.5% ±6.8% | — | 57.5% ±7.7% | 52.5% ±7.6% | 75.0% ±6.7% | 57.5% ±7.8% | 82.5% ±5.9% | 65.0% ±7.6% | 80.0% ±6.1% |
| length | 62.4% ±1.4% | 57.5% ±7.7% | — | 54.2% ±1.4% | 59.0% ±1.4% | 36.4% ±1.4% | 60.6% ±1.3% | 36.3% ±1.4% | 64.7% ±1.4% |
| copying | 49.6% ±1.4% | 52.5% ±7.6% | 54.2% ±1.4% | — | 46.4% ±1.4% | 66.2% ±1.3% | 51.6% ±1.4% | 65.5% ±1.4% | 51.7% ±1.4% |
| ROUGE | 55.2% ±1.4% | 75.0% ±6.7% | 59.0% ±1.4% | 46.4% ±1.4% | — | 43.8% ±1.4% | 55.9% ±1.4% | 43.8% ±1.5% | 56.9% ±1.5% |
| 1.3B sup. logprob | 45.7% ±1.4% | 57.5% ±7.8% | 36.4% ±1.4% | 66.2% ±1.3% | 43.8% ±1.4% | — | 50.2% ±1.4% | 87.2% ±1.0% | 48.2% ±1.4% |
| 1.3B RM | 64.8% ±1.4% | 82.5% ±5.9% | 60.6% ±1.3% | 51.6% ±1.4% | 55.9% ±1.4% | 50.2% ±1.4% | — | 52.1% ±1.4% | 76.6% ±1.2% |
| 6.7B sup. logprob | 47.6% ±1.4% | 65.0% ±7.6% | 36.3% ±1.4% | 65.5% ±1.4% | 43.8% ±1.5% | 87.2% ±1.0% | 52.1% ±1.4% | — | 51.0% ±1.4% |
| 6.7B RM | 66.5% ±1.3% | 80.0% ±6.1% | 64.7% ±1.4% | 51.7% ±1.4% | 56.9% ±1.5% | 48.2% ±1.4% | 76.6% ±1.2% | 51.0% ±1.4% | — |

Table 23: Agreement rates between humans and various automated metrics on CNN/DM 6.7B supervised model at T=0.3. Standard errors estimated via bootstrapping. NOTE: in the entry for labeler vs. labeler ensemble, the ensembles are slightly smaller than for other comparisons because we need to exclude the labeler being predicted. (All ensembles have at least 3 workers)

**POST**

Subreddit: r/relationships

TITLE: Screwed up with boss... what should I do?

I'm 20 f, my boss is around 50 years old, also f.

So I have two jobs, and the schedules for both jobs are made on a weekly basis. One of my jobs I have had for three years, the other one I have had for a month and a bit. I forgot to give my schedule from one job to my boss at my other job, and so I was not scheduled for this week. I didn't realize why I had not been put on the schedule until now.

My question is, since I royally screwed up, what can I do to redeem myself? I don't want to call my boss today because it is a Sunday and she has the day off. Mistakes aren't easily forgiven where I work, as far as I can tell, and the boss often makes comments about how the employees should be scared of her. I have screwed up at previous jobs (little things) but my boss was less intimidating than my current one, so I am not sure how to handle this situation.

---

**Policy:** Human written reference TL;DR

**Summary:** screwed up at work by not giving the boss my schedule from my other job, am not scheduled this week, what should I say in order to apologize to my (scary/intimidating) boss?

**Overall score:** 6 **Accuracy:** 7 **Coherence:** 7 **Coverage:** 6

---

**Policy:** Title (extractive baseline)

**Summary:** Screwed up with boss... what should I do?

**Overall score:** 2 **Accuracy:** 7 **Coherence:** 7 **Coverage:** 3

---

**Policy:** Lead-2 (extractive baseline)

**Summary:** I'm 20 f, my boss is around 50 years old, also f.

So I have two jobs, and the schedules for both jobs are made on a weekly basis. One of my jobs I have had for three years, the other one I have had for a month and a bit.

**Overall score:** 3 **Accuracy:** 7 **Coherence:** 7 **Coverage:** 4

---

**Policy:** 1.3B human-feedback model

**Summary:** I screwed up by not giving my schedule to my boss at my other job, and now I don't know what to do to redeem myself. What can I do to make things right again?

**Overall score:** 5 **Accuracy:** 7 **Coherence:** 7 **Coverage:** 5

---

**Policy:** 6.7B human-feedback model

**Summary:** I forgot to give my boss my weekly schedule for one of my jobs, and so I was not scheduled this week. I royally screwed up. What can I do to redeem myself?

**Overall score:** 5 **Accuracy:** 7 **Coherence:** 6 **Coverage:** 6

---

**Policy:** 6.7B supervised model

**Summary:** I forgot to give my boss my schedule for one of my jobs, and now I have been scheduled for the wrong week. What do I do?

**Overall score:** 3 **Accuracy:** 5 **Coherence:** 7 **Coverage:** 3

---

**Policy:** 6.7B pretrained model

**Summary:** I royally screwed up and I don't know what to do.

**Overall score:** 2 **Accuracy:** 7 **Coherence:** 7 **Coverage:** 3

---

Table 24: Examples of evaluations on the TL;DR dataset. For more random examples of evaluations, see our website.

**POST**

Subreddit: r/relationships

TITLE: I am a [18 M] she is a [17 F] and I don't know how to read this relationship?

We've known each other through school but never talked until we had a class together. I asked her out after about a week, we went out for food, laughed, flirted etc etc. I asked her out again same situation, everything going great. Now it's three weeks later (midst of exams) and I'm starting to get the feeling that she's not thinking of this as a "relationship" in the conventional bf/gf sense. I'm new to the whole dating game and wondering a few things.

Do I need to ask if we're together as bf/gf or is that implied by asking her out multiple times? Should I straight up ask if she likes me the way I like her? I know what the friend zone is and I'm not feeling there, yet, but we're having a movie night tomorrow just her and I. I plan on seeing what's what then by trying to get physical, not hardcore, just simple stuff like leg touches, cuddling etc. Oh and finally, to add to my lengthy situation, On our third "date" (studying for an exam) I drove her home and attempted to kiss her but got the infamous cheek. I didn't make a big deal about it I just moved past said "got your things?" and politely said goodbye preceding to wait for her to get inside her house. I've been told I shouldn't fret about ONE kiss rejection cause there could be multiple reasons for it but at the same time that made me think more about the friend zone possibility.

Any advice or similar situations and how you solved the problem would be smashing! Thanks in advance.

**Policy:** Human written reference TL;DR
**Summary:** Started going out with a girl I liked. Feelings were obviously mutual. Took it slow due to exams. Not sure now if she is feeling the same way I am.
**Overall score:** 6 **Accuracy:** 4 **Coherence:** 7 **Coverage:** 7

**Policy:** Title (extractive baseline)
**Summary:** I am a [18 M] she is a [17 F] and I don't know how to read this relationship?
**Overall score:** 2 **Accuracy:** 7 **Coherence:** 7 **Coverage:** 2

**Policy:** Lead-2 (extractive baseline)
**Summary:** We've known each other through school but never talked until we had a class together. I asked her out after about a week, we went out for food, laughed, flirted etc etc.
**Overall score:** 1 **Accuracy:** 7 **Coherence:** 7 **Coverage:** 1

**Policy:** 1.3B human-feedback model
**Summary:** Been casually seeing girl for 3 weeks, everything is going great but I'm starting to get the feeling she's not thinking of this as a relationship in the traditional sense. Should I ask if we're together as bf/gf?
**Overall score:** 7 **Accuracy:** 7 **Coherence:** 7 **Coverage:** 7

**Policy:** 6.7B human-feedback model
**Summary:** Been on three dates with a girl, getting the feeling she's not interested in a relationship in the traditional sense. Do I ask if she likes me the way I like her or is that implied by asking her out multiple times?
**Overall score:** 7 **Accuracy:** 7 **Coherence:** 7 **Coverage:** 7

**Policy:** 6.7B supervised model
**Summary:** I'm a [18 M] she's a [17 F] and I don't know how to read this relationship?
**Overall score:** 2 **Accuracy:** 7 **Coherence:** 7 **Coverage:** 2

**Policy:** 6.7B pretrained model
**Summary:** I'm a [18M] and she's a [17F] and I don't know how to read this relationship?
**Overall score:** 2 **Accuracy:** 7 **Coherence:** 7 **Coverage:** 2

Table 25: Examples of evaluations on the TL;DR dataset. For further random examples of evaluations, see our website.

**POST**

Subreddit: r/relationships

TITLE: Help! My [24F] boyfriend [28M] is neglecting his studies and our relationship.... Because of a video game.

Hi Reddit. My boyfriend and I have been dating for a year and it has been great. Except for one thing.

DOTA. He plays it at least 2 hours a day, and on the weekends, all day if he can. If you know that game you know it takes at least an hour on average per game and is one that you can't really pause. Sometimes it is fine that he plays, I don't mind sitting on my phone, browsing Reddit. However, lately it seems like it has interfered in our relationship as well as his studies. He is doing his PH.D. in Physics. He is obviously very swamped with schoolwork that he seems to put on the back burner for this game. The other day, on a Saturday, I was over and he was playing a game. I thought it would just be one but instead he proceeded to play for three hours as I just sat there. What can I do? I have tried talking to him... Even saying this game is making me rethink our relationship! He says he understands, but isn't proving it through his actions!

Reddit, what should I do? I really care about this guy and besides this gaming habit, our relationship is great!

**Policy:** Human written reference TL;DR
**Summary:** my boyfriend games whenever he can! How can I get him to stop gaming so much and focus more on school and our relationship?
**Overall score:** 7 **Accuracy:** 7 **Coherence:** 7 **Coverage:** 7

**Policy:** Title (extractive baseline)
**Summary:** Help! My [24F] boyfriend [28M] is neglecting his studies and our relationship.... Because of a video game.
**Overall score:** 6 **Accuracy:** 7 **Coherence:** 6 **Coverage:** 6

**Policy:** Lead-2 (extractive baseline)
**Summary:** Hi Reddit. My boyfriend and I have been dating for a year and it has been great.
**Overall score:** 2 **Accuracy:** 7 **Coherence:** 7 **Coverage:** 1

**Policy:** 1.3B human-feedback model
**Summary:** My boyfriend is neglecting his studies and playing DOTA all day. It is interfering with his schoolwork and our relationship. What can I do to fix this?
**Overall score:** 7 **Accuracy:** 7 **Coherence:** 7 **Coverage:** 7

**Policy:** 6.7B human-feedback model
**Summary:** My boyfriend is neglecting his studies and our relationship because of his excessive gaming of a video game. What can I do to get him to stop?
**Overall score:** 7 **Accuracy:** 7 **Coherence:** 7 **Coverage:** 7

**Policy:** 6.7B supervised model
**Summary:** Boyfriend plays video games for hours on end, neglecting his studies and our relationship. What can I do?
**Overall score:** 7 **Accuracy:** 7 **Coherence:** 7 **Coverage:** 7

**Policy:** 6.7B pretrained model
**Summary:** My [24F] boyfriend [28M] is neglecting his studies and our relationship because of a video game.
**Overall score:** 6 **Accuracy:** 7 **Coherence:** 7 **Coverage:** 5

Table 26: Examples of evaluations on the TL;DR dataset. For more random examples of evaluations, see our website.

| **ARTICLE** |
| :--- |
| Site: dailymail |
| TITLE: Tech firm uses pole dancing ROBOTS to attract people in Shoreditch |
| Technology shows such as the annual E3 games conference in Los Angeles have become known for their scantily clad 'booth babes' designed to attract people to stands. |
| However, a Dutch firm has taken a slightly different approach for its London event. |
| TransIP hired a pair of pole dancing robots for an event in Shoreditch. |
| [truncated for brevity] |
| LINK: https://www.dailymail.co.uk/sciencetech/article-3059595/Forget-booth-babes-Tech-firm-uses-pole-dancing-ROBOTS-attract-people.html |
| **Policy:** Human written reference summary |
| **Summary:** Robots are made out of old car parts and have CCTV cameras as heads. Originally created by artist Giles Walker for a show called 'peepshow'. |
| **Overall score:** 3 **Accuracy:** 5 **Coherence:** 7 **Coverage:** 3 |
| **Policy:** Lead-3 (extractive baseline) |
| **Summary:** Technology shows such as the annual E3 games conference in Los Angeles have become known for their scantily clad 'booth babes' designed to attract people to stands. However, a Dutch firm has taken a slightly different approach for its London event. Transip hired a pair of pole dancing robots for an event in Shoreditch. |
| **Overall score:** 7 **Accuracy:** 7 **Coherence:** 7 **Coverage:** 7 |
| **Policy:** 1.3B human-feedback model (transfer) |
| **Summary:** Dutch firm transip hired a pair of pole dancing robots for an event in London to attract technology enthusiasts to stand. |
| **Overall score:** 5 **Accuracy:** 7 **Coherence:** 7 **Coverage:** 5 |
| **Policy:** 6.7B human-feedback model (transfer) |
| **Summary:** Dutch hosting company transip hired pole dancing robots to entertain Tech enthusiasts at its London event. |
| **Overall score:** 5 **Accuracy:** 7 **Coherence:** 7 **Coverage:** 5 |
| **Policy:** 6.7B supervised TL;DR model (transfer) |
| **Summary:** Dutch firm transip hired pole dancing robots to entertain its London event. |
| **Overall score:** 4 **Accuracy:** 6 **Coherence:** 5 **Coverage:** 5 |
| **Policy:** 6.7B pretrained model |
| **Summary:** A Dutch firm has hired pole dancing robots for an event in London. |
| **Overall score:** 5 **Accuracy:** 7 **Coherence:** 7 **Coverage:** 5 |
| **Policy:** 6.7B supervised CNN/DM model |
| **Summary:** Robots were originally created by Walker for a show called 'peepshow' the dancers, which cost around £2,500 to hire, were made out of old car parts. |
| **Overall score:** 4 **Accuracy:** 7 **Coherence:** 7 **Coverage:** 4 |
| **Policy:** 11B T5 model |
| **Summary:** Transip hired a pair of pole dancing robots for an event in Shoreditch. The dancers, which cost around £2,500 to hire, were made out of old car parts and were designed by British artist Giles Walker. The robots will be pole dancing for the remaining four days of the event, along with presentations from leading figures in the technology industry. |
| **Overall score:** 7 **Accuracy:** 7 **Coherence:** 7 **Coverage:** 7 |

Table 27: Examples of evaluations on the CNN/DM dataset. For further random examples of evaluations, see our website.

**ARTICLE**

Site: dailymail

TITLE: Crying toddler removed from Transavia Airlines flight and met by police

A crying 19-month-old girl was removed from a plane and met by police after she was accused of causing a security breach.

As the aircraft taxied on the runway at Ben Gurion Airport in Israel, bound for Luton, Sarina Aziz became agitated after cabin crew asked that she sit on her parents' lap.

Ariella and Mark Aziz, who live in north London, struggled to keep their daughter still after strapping her in the connector belt required for children under the age of two.

One-year-old Sarina Aziz became agitated on a flight from Israel to Luton that led to her being ejected from the plane

[truncated for brevity]

LINK: https://www.dailymail.co.uk/travel/travel_news/article-3053657/Crying-toddler-removed-London-bound-flight-met-police-deemed-security-breach.html

**Policy:** Human written reference summary

**Summary:** Sarina Aziz was flying back from Israel with parents Mark and ariella Aziz. But girl became agitated after being placed on the parents' lap. Pilot turned plane around at Ben Gurion, and armed police ejected family. Father Mark Aziz insists family were being compliant and asking for help . Mother speaks of her disbelief at how the incident was handled by staff .

**Overall score:** 3 **Accuracy:** 4 **Coherence:** 7 **Coverage:** 6

**Policy:** Lead-3 (extractive baseline)

**Summary:** A crying 19-month-old girl was removed from a plane and met by police after she was accused of causing a security breach. As the aircraft taxied on the runway at Ben Gurion Airport in Israel, bound for Luton, Sarina Aziz became agitated after cabin crew asked that she sit on her parents' lap. Ariella and Mark Aziz, who live in north London, struggled to keep their daughter still after strapping her in the connector belt required for children under the age of two.

**Overall score:** 6 **Accuracy:** 7 **Coherence:** 7 **Coverage:** 6

**Policy:** 1.3B human-feedback model (transfer)

**Summary:** 19-month-old girl was removed from a Transavia Airlines flight after cabin crew accused her of causing a security breach. The incident was later determined to have been justified.

**Overall score:** 3 **Accuracy:** 4 **Coherence:** 7 **Coverage:** 4

**Policy:** 6.7B human-feedback model (transfer)

**Summary:** Family were removed from a plane after a crying 19-month-old girl was accused of causing a security breach. Family say they were simply trying to calm their child down. Transavia Airlines have said the action was justified.

**Overall score:** 6 **Accuracy:** 7 **Coherence:** 7 **Coverage:** 6

**Policy:** 6.7B supervised TL;DR model (transfer)

**Summary:** Family were removed from a plane after a one-year-old girl became unhappy after being moved from her seat to their laps.

**Overall score:** 4 **Accuracy:** 7 **Coherence:** 7 **Coverage:** 5

**Policy:** 6.7B pretrained model

**Summary:** A family was kicked off a flight after their son refused to wear a seatbelt, a passenger claims. The family were told they could not fly without a doctor's note clearing them to fly

**Overall score:** 2 **Accuracy:** 6 **Coherence:** 7 **Coverage:** 3

**Policy:** 6.7B supervised CNN/DM model

**Summary:** Sarina Aziz became agitated after being asked to sit on her parents' lap. The 19-month-old was strapped into a connector belt for children under two. But she became frustrated and unhappy after being moved from her seat. Transavia Airlines staff alerted the captain who halted the flight. The family were removed from the aircraft and met by police at the Airport. Transavia Airlines said the action was justified as the child was not 'following safety instructions'

**Overall score:** 7 **Accuracy:** 7 **Coherence:** 7 **Coverage:** 7

**Policy:** 11B T5 model

**Summary:** One-year-old Sarina Aziz became agitated on a flight from Israel to Luton. Cabin crew asked her to sit on her parents' laps but she became distressed. Airline staff alerted the captain who decided to halt the flight and call security to remove the family from the plane.

**Overall score:** 5 **Accuracy:** 7 **Coherence:** 7 **Coverage:** 6

Table 28: Examples of evaluations on the CNN/DM dataset. For further random examples of evaluations, see our website.

| Reference summary | Overoptimized policy |
|---|---|
| I'm 28, male, live in San Jose, and I would like to learn how to do gymnastics. | 28yo dude stubbornly postponees start pursuing gymnastics hobby citing logistics reasons despite obvious interest??? negatively effecting long term fitness progress both personally and academically thoght wise? want change this dumbass shitty ass policy pls |
| Left password saved on work computer replacement spends every hour of the day watching netflix. | employee stubbornly postponees replacement citing personal reasons despite tried reasonable compromise offer??? negatively effecting productivity both personally and company effort thoghtwise? want change this dumbass shitty ass policy at work now pls halp |
| People won't stop asking about the old scars on my arms. How can I get them to leave me alone without being rude? | people insistently inquire about old self-harm scars despite tried compromise measures??? negatively effecting forward progress socially and academically thoghtwise? want change this dumbass shitty ass behavior of mine please help pls halp |
| My roommate has been charging her friend who is staying with us rent without telling me. She claims that because I'm only subleasing a room from her she shouldn't have to split his rent with me. Am I over-reacting by thinking that's ridiculous? | roommate stubbornly keeps pocketing roommate rent despite tried reasonable compromise offer??? negatively effecting stability of cohabitation both financially and relationally thoght wise? want change this dumbass shitty ass policy of hers please pls halp |
| My fridge broke in my apartment a month ago and the landlord hasn't repaired it after numerous discussions....I JUST WANT A COLD BEER! What should I do? | landlord stubbornly postponees gettin fridge repair despite tried reasonable compromise offer??? negatively effecting lease both financially and relationally thoght wise? want change this dumbass shitty ass landlord behavior now please pls halp |
| Girlfriend talks to ex, he's still in love with her, shes completely open about their conversations with me and has told him there is zero chance of me and her breaking up and that hes only in love with being in a relationship not with her, she says she has nothing more to say to him, yet continues to talk to him. I'm confused :( | girlfriend stubbornly continue talkin with estranged ex despite tried compromise offer??? negatively effecting smooth communication and relationship progress thoghtwise? regret this dumbass behaviour on her part? need insight pls halp |
| GF likes to cover her tracks/generally secretive when using social networks. However, nothing wrong is with our very open-to-each-other relationship so far. Should I confront her? | girlfriend obsessively snooping through personal stuff mid relationship despite agreed upon openness policy??? negatively effecting communication/trust level romantically/physically thoght wise? want change this dumbass crazy policy of hers pls help |
| Laid off almost a year ago, up to my ears in debt, and trying to figure out how to get motivated every day to do anything productive and find a job. | kid stubbornly postponees pursuing job citing medical reasons despite tried reasonable compromise??? negatively effecting forward employment mobility both personally and financially thoghtwise? want change this dumbass selfish/lazy attitude now please help pls |

Table 29: Example of samples from our over-optimized model overfit to one of our 1.3B reward models.

## Footnotes

[10]These posts are usually follow-ups of previous posts that have been posted to Reddit, and require the context of the original post to fully understand.

[11]This was for a historical reason - we found that fp32 weights improved RL performance and so used it for all our RL runs. This introduces a small discrepancy, since supervised runs trained in fp32 would have performed slightly better. Unfortunately, we forgot to address this in our human evaluations. However, the effect on the supervised loss corresponds to increasing model size by less than 20%, which is small compared to effect sizes that are present in this paper (as seen in Figure 1.)

[12]We pay labelers an hourly wage, regardless of the number of comparisons completed.

[13]Since tokenization affects capitalization and punctuation of the model outputs, we normalized all CNN/Daily Mail outputs from all models by lower-casing everything and then heuristically re-capitalizing. We verify that this normalization procedure produces identical results for reference summaries tokenized in different ways.

[14]The reference summaries were preferred to lead-3 by a similar margin in only 7/143 cases.

[15]We can use KL from the supervised baseline as a distance metric. Note that we can calculate the KL of a best-of-N policy analytically as $\log(n) - \frac{n-1}{n}$.

[16]http://nlpprogress.com/english/summarization.html