[Reviews · NeurIPS 2020]

Review 1

Summary and Contributions: This paper presents a thorough study on text summarization from human feedbacks. The motivation is that both likelihood training and overlap-based scores (e.g., BLEU and ROUGE) are bad at modeling real human satisfaction. The paper first pre-trains a model with MLE against reference summaries and then reinforces towards the human feedback by PPO, where the reward is a learned binary classifier. The results show that the approach is better than supervised learning as well as reference summaries (which are usually noisy, like TL;DR, or headline).

Strengths: The experiments are very well conducted and documented in the paper and 27-page sup.

Weaknesses: However, I have two major concerns: 1. As also mentioned by the authors, this paper is basically an expanded analysis of [3, 58]. Basically, the key techniques of classification-based reward and PPO have been explored in [58], and the major extension is that this paper uses a larger and better-engineered model, and adapts an online setting to the offline setting. Therefore, I feel this paper has very little novelty in the sense of machine learning. The authors are very honest about this in the Related Work (Line 86), though. However, the Border Impacts section should be disentangled with [3, 58]. I feel the contribution of this paper mainly lies in the engineering for the summarization task and a documentation of extensive experiments. 2. The paper claims ROUGE scores may not be good for measuring human satisfaction (Line 4). While this is intuitive, it is not tested in this paper. Suppose we perform RL towards ROUGE scores and achieve similar performance, then there's no point in doing RL by human feedback. In fact, the authors report that their model achieves high performance in 1--7 Likert human evaluation (main paper) as well as ROUGE scores (Sec F.4). Isn't it showing ROUGE actually highly correlates with human satisfaction? The author claims offline human feedback is better than online human feedback. But is this tested? Other concerns: 3. The formulation of Sec 3.4 -> "Reward models" is a little bit confusing. The reward model r_theta takes as input a sentence x and a candidate summary y, and outputs if this summary is better. But since the reward model r_theta only considers one candidate summary at a time, it's unable to predict if a summary is *better* or not (which is only meaningful when comparing two summaries). I feel a better way of formulation is that the reward model r_theta predicts if a summary is *good* or not. In pairwise human annotation, a higher-scored summary is deemed good, whereas a lower-scored summary is deemed bad. Then, it's unsure if the human feedback protocol used in this paper is optimal. Maybe we can do pointwise annotation. 4. The results show that lead-3 baseline (I guess it's choosing the first three words? Or tokens?) is better than the model by 0.5-1 point in 1--7 rating on the CNN/DM dataset. Does it mean that the learned models are not effective at all on this dataset, or this dataset has systematic issues preventing it from serious research at this stage? Since this paper has significant experimenting efforts, this paper probably worth publishing (compared with other apply-A-to-B papers).

Correctness: Yes, the experiments appear to be very rigorous and well documented.

Clarity: Yes.

Relation to Prior Work: Yes.

Reproducibility: Yes

Additional Feedback: UPDATE: I am raising 1 point (6->7) because the authors well addressed one of my major concerns. My confidence score is unchanged, because I am not 100% sure if this paper would attract broad enough audience in NeurIPS compared with an NLP conference (say EMNLP, which has almost the same timeline as NeurIPS). However, I do not view it as a major concern. NeurIPS allows machine learning applications papers, and this paper does add to the evidence of human-in-the-loop learning in general. Additionally, the border between ML and NLP isn't that clear and should not be the main scientific judgment. While some reviewers have concerns on the writing, I would mainly judge a paper by its substance rather than formality. In addition, I think the current writing style is an ADVANTAGE rather than disadvantage. The current writing helps easily understand the main idea of the paper, and if interested, audience can look into supplemenary materials for more details or additional evidence. It's actually much easier to read than a tedious, book-long journal paper (which oftentimes is repetitive in writing). I do have one suggestion on writing, though. Maybe you can include a table of content for appendix, listing the title of each appendix section and optionally summarizing the main findings in that section.


Review 2

Summary and Contributions: This paper explores using RL (PPO) to learn an abstractive summarization model from human feedback. Humans are presented with ground truth references and samples from the model and asked to select which they prefer. This is then used to train a reward model, which is combined with a KL term keeping the summarization model close to a version fine-tuned with basic supervised learning. This procedure can be done iteratively: better models can generate better candidates for human preference evaluation. The paper shows that on the TL;DR corpus, training large GPT transformer models (1.3B-6.7B) with human rewards leads to a model that actually outperforms the human references as rated by humans, which standard supervised learning fails to do. The paper analyzes the factors that lead to these summaries being better, and finds that coverage of important concepts is one. In addition, Figure 5 shows that it is important to balance the reward learning with distance from the supervised baseline (optimizing too heavily for human judgments and straying too far from the base supervised model starts to hurt performance). Finally, the paper studies transfer to CNN/DM and finds strong results there from human feedback as well, suggesting these preferences may be more transferable than supervised examples.

Strengths: This paper has two particularly notable contributions (neither being the technique, which is very similar to that of prior work). First, the paper scales up the human preference learning both in terms of underlying model size as well as data collection. As a result, the system they build is extremely strong: it performs admirably well in comparisons against human references on both reddit and CNN/DM. The conclusions are therefore probably substantially different from those of other work in this vein, including the prior Ziegler work, and are notable as a result. Second, the dataset produced is likely to be of significant utility to other summarization researchers interested in human judgments on this task (which should be many of them :) ). The authors have collected a lot of data, although it was actually difficult for me to determine the total number of judgments -- do I have to sum the last column of Table 10 in the appendix? It seems like this is at least 64k given Figure 6. Empirically this paper holds itself to an extremely high standard. The website showing non-cherry-picked examples is great and was very fun to play with and informative. The human evaluation is done carefully. The appendices are good evidence of this. The authors have put substantial thought into how to train annotators and prompt them to do the task. The comparison between researcher agreement rates and labeler agreement rates is also useful for calibrating the quality of the labeling. Overall, the authors have extensively shown that their model works very well, and it is likely to set a new standard for this task.

Weaknesses: This paper essentially presents nothing "novel." The authors freely admit that they have taken the approach of Ziegler et al., made minor modifications, and scaled it up. However, given the significance of the empirical results and the additional data collection effort, I do not think this is a dealbreaker for NeurIPS. Unfortunately, the paper's system is also unattainable to many researchers. Optimizing their 6.7B model against human preferences took 320 GPU days, and this was an incremental process so presumably was done many times throughout development. However, this is simply a fact of life in pre-training research in 2020. Most critically, the paper doesn't really have too much comparison to techniques from other work. The paper's baselines are worthy: they use a supervised baseline and a T5-fine-tuned model. However, given the massive amounts of work in the summarization/pre-training space (particularly PEGASUS), it would've been nice to see more empirical comparison with other techniques and understand how these stack up in absolute terms. At the very least, the authors should add some discussion about relevant prior work in summarization beyond reward learning from human judgments: what other techniques promise to improve factors like coverage?

Correctness: This work is very carefully done from an empirical standpoint. I am convinced by the paper's claims.

Clarity: Except for my inability to determine exactly how many judgments were collected, the paper was mostly clear. It's pretty easy to read as many of the details are delegated to the appendix; this does lead to lots of cross-references but makes the main text clear to follow.

Relation to Prior Work: Generally yes; as discussed above, some more empirical comparisons could be nice to see. But I appreciate that the authors are up front about the non-novelty of their approach.

Reproducibility: Yes

Additional Feedback: ============== Post-response: Thanks for the response. My assessment of the paper is unchanged, though I have upgraded my confidence in the score.


Review 3

Summary and Contributions: This paper presents a summarization model by fine-tuning large pre-trained models based on rewards learned from pairwise human preference. The trained summarizer is able to produce summaries preferred by human over references (which are also written by human).

Strengths: + The framework shows that neural models can pick up human reference in terms of summary quality (which could be hard to be quantified, e.g. compared to lengths).

Weaknesses: - The fine-tuning process takes long time to finish. The cross-domain experiments on news show that in-domain tuning is still necessary. - Importantly, as the authors notice, the reward functions can pick up minor details, which means it can be sensitive to the change of summaries, which can be about content or grammar. Given that the reward function is key for model training, it is thus necessary to give more analysis on how the quality and diversity of the training pairs would affect reward function learning.

Correctness: The implementation seems to be correct to me.

Clarity: The paper is mostly written well, with several places that more details can be used. The paper often refers to results in appendix, hurting comprehension in general. It's like I'm reading a list of conclusions with results presented at the right place.

Relation to Prior Work: The related work has reflected the recent advances in the area.

Reproducibility: Yes

Additional Feedback: Update: I thank the authors for their feedback. I think this work would benefit from including the main results at the right places in the main paper, rather than listed in a lengthy supplementary file. An NLP journal paper would be more proper in this case. The major contribution of this work lies in the analysis of experimental results based on the models adopted from prior work. There are two major directions this work can improve on. First, analyzing which types of human feedback lead to what improvement. E.g. line 250 mentioned "make minimal edits", this can be clarified and categorized into different types of edits. Second, I'm not sure if 8-page is enough for this type of work, since many results really should be presented in this main paper.


Review 4

Summary and Contributions: The authors collect a large, high-quality dataset of human summary evaluations. They also use that dataset to fine-tune pre-trained summarization models with reinforcement learning. They experiment on the Reddit TL;DR dataset and significantly outperform both human references and all the baselines.

Strengths: The authors collect a high-quality summarization dataset. This would be useful for directly optimize the summarization model with the human feedback score.

Weaknesses: The paper is not well organized. It is hard for me to follow the details, contributions, and experimental results. My recommendation is to carefully reference other well-structured papers, such as DeepSets (https://arxiv.org/abs/1703.06114), and SoftmaxBottleneck(https://arxiv.org/abs/1711.03953). Followings are some of my recommendations to improve readability. - The title "Learning to summarize from human feedback" is vague. There exist other works also use human feedback to enhance the quality of summarization. It is better to specify the differences compared to existing works. - Many parts of the paper depend on the Appendix. For example, in section 3.4 Models, most of the details are not specified in the main paper. Conference papers must be understandable without the Appendix. - In the introduction section, I recommend the authors to specify the main contributions. For example, from the SoftmaxBottleneck paper, the authors finish the introduction section with "Our contributions are two-fold. First, we identify the Softmax bottleneck by ... Second, we propose a simple and effective method ...". - Paper [58] must be added to the experiments section. As the authors mentioned, [58] seems to be the most related baseline method. It is better to specify what is and how much improved compared to [58]. The authors mentioned their improvements in line #85~#88 and #142~#146, but most are vague and not persuasive. For example, "maintain high-touch relationship with labelers" cannot be a good contribution.

Correctness: Not enough. Many details are omitted.

Clarity: Not enough. See "Weakness" section for my recommendations.

Relation to Prior Work: Not enough. See "Weakness" section for my recommendations.

Reproducibility: Yes

Additional Feedback: Post AR: Thank you for preparing the author response and apologies for my misunderstanding. I'm raising my score by four points (2->6) as I completely underestimated this work. I believe, with new checkpoints and datasets, many interesting follow-up works will come out. I still think journal format is adequate to send messages clearly (like T5 paper), but I'm on the acceptance side due to impressive results. =======================================================

[Author Response · NeurIPS 2020]

We thank the reviewers for their comments. We are pleased that reviewers found the paper to be very empirically
convincing (**R1**, **R2**), that the experiments are rigorous and well-documented (**R1**, **R2**), and that the human feedback
dataset we collect would be useful for the community (**R2**, **R4**). We respond to the concerns raised by reviewers below.

One of the primary concerns was that our paper lacks novelty relative to prior work. We agree: novelty is not the aim of
the paper. Rather, the strength of the paper is in obtaining extremely strong empirical results using a technique that has
received relatively little attention, on an important task (abstractive summarization). **R2** agrees that it is likely to set a
new standard for the task. Specifically, our main contributions are:

**(1) We show that training with human feedback significantly outperforms very strong baselines** (human-written
reference summaries and large supervised models). To do this, we: (a) scale up model size, (b) move data collection to
the batch setting, closely monitoring human data quality, and (c) separate the value and policy networks.

**(2) We show human feedback models generalize much better to new domains than supervised models.** Our
TL;DR model matches human reference summaries on CNN/DM news articles without any fine-tuning, while analogous
supervised baselines trained on only TL;DR regress to the performance level of pretrained baselines.

**(3) We conduct extensive empirical analyses** of both our reward model (effect of data/model size, over-optimization,
what it has learned), and of other metrics such as ROUGE (predictiveness of human judgements, over-optimization).

**(4) We collect a large dataset** of human comparisons of summaries, which we aim to release publicly.

We agree with **R4**'s suggestion that we reframe the introduction to highlight these contributions and will update this.

@**R2**, **R4**: **Comparing to more baselines (e.g. PEGASUS or Ziegler et al.).** We agree this would be ideal, but
comparing to other works using human preferences is expensive. We will follow **R2**'s recommendation to add more
discussion of prior work (including PEGASUS) in the final version. Note that Table 3 of the Pegasus paper suggests
they underperform compared to reference summaries on the Reddit TIFU dataset, while we convincingly outperform
reference summaries on our Reddit dataset (including the TIFU subset). We believe that comparing to prior work with
much smaller models is uninformative and masks gains from the technique. We agree with **R2**'s assessment that our
chosen baselines (human-written reference summaries, large supervised models, T5) are strong.

@**R1**: **The paper doesn't test if ROUGE is good for measuring human preferences.** These results are detailed in
the appendix. **R1** states "if RL on ROUGE achieves similar performance, there is no point of human feedback", but in
Figure 13 we find optimizing ROUGE performs significantly worse than optimizing our RMs, with "over-optimization"
after just a few bits of pressure. **R1** states "our model achieves high ROUGE scores". However, this does not imply
high correlation with human preferences, which depends on the sample distribution. In Appendix F.7, we find ROUGE
and human preferences to be correlated for supervised baselines, but uncorrelated for samples from our best model,
providing more evidence that as models improve, ROUGE stops tracking quality. Similarly, in Figure 14a human
feedback models achieve *lower* ROUGE scores than the supervised baselines on TL;DR.

@**R4**: **"Maintaining a very close feedback loop with labelers cannot be a good contribution."** We disagree. While
we understand that human data collection techniques are difficult to ablate, our view is that the standard method of
collecting human data in the field — using crowdsourced websites with minimal researcher-labeler interaction — is a
significant impediment to using this data for training models (as evidenced by Ziegler et al.). We believe our processes
(onboarding, monitoring agreement rates, providing feedback, etc. see Appendix C.1) were important for strong results.
We hope our results convince researchers to pay closer attention when collecting human data.

@**R1** **"Maybe we can do pointwise annotation".** This can work but there will be discrepancies across labelers and
drift while shifting distributions. Comparisons also let us measure worker agreement which we use to control quality.

@**R3**: **"The cross-domain experiments on news show that in-domain tuning is still necessary."** Figure 4 shows
the opposite: our models achieve very strong performance on news with no in-domain training. In-domain supervised
fine-tuning slightly improves quality scores but this is explained by the longer length of CNN/DM summaries. We
encourage reviewers to inspect the non-cherry-picked samples themselves.

@**R4**: **"The paper is not well organized."** We believe this may partially be because we present our contributions as
"results" in the introduction, as this is an empirical paper. We are open to specific suggestions.

@**R3**, **R4**: **Too many details in Appendix.** As **R3** notes, we struggled with the page limit, and interesting details (e.g.
Figure 13) are omitted from the main text. Opinion on this is perhaps largely personal preference. **R2** enjoyed having
details in the appendix. **R4** prefers model details to be in the main text, but we disagree as our models are standard.

@**R1**: **What does it mean that lead-3 outperforms the reference summaries on CNN/DM?** We were also surprised
by this result, which deserves further attention. Appendix D contains some more detailed analysis.

[Meta-Review · NeurIPS 2020]

This paper presents an empirical study on learning summarization models from human feedback. The authors use RL (PPO) to learn an abstractive summarization model from human judgements on top of an MLE-based supervised model. The thorough experiments produce strong results in the large-scale and cross-domain settings. Cons: As the authors admitted, this paper essentially has little technical novelty compared to previous existing works. Pros: However, the paper obtains strong empirical results with large data scale and model size, which is likely to set a new standard for the task of summarization, and the human feedback dataset collected with the experiments is likely to be very useful for other researchers in this area. Although all reviewers acknowledge the merit of this empirical paper, there are complaints about the long appendix and referring too much to it in the main paper. We hope the authors will take this into consideration and include important results from the appendix. Based on these, our recommendation for this paper is acceptance.